# *SePer*: Measure retrieval utility through the lens of semantic perplexity reduction

**Lu Dai[2,1], Yijie Xu[1], Jinhui Ye[1,3], Hao Liu[1,2,\*], Hui Xiong[1,2,\*]**
[1]The Hong Kong University of Science and Technology (Guangzhou), Guangzhou, China
[2]The Hong Kong University of Science and Technology, Hong Kong SAR, China
[3]Carnegie Mellon University, Pittsburgh, USA
`ldaiae@connect.ust.hk, yxu409@connect.hkust-gz.edu.cn`
`jinhuiy@andrew.cmu.edu, liuh@ust.hk, xionghui@ust.hk`

## Abstract

Large Language Models (LLMs) have demonstrated improved generation performance by incorporating externally retrieved knowledge, a process known as retrieval-augmented generation (RAG). Despite the potential of this approach, existing studies evaluate RAG effectiveness by 1) assessing retrieval and generation components jointly, which obscures retrieval's distinct contribution, or 2) examining retrievers using traditional metrics such as NDCG, which creates a gap in understanding retrieval's true utility in the overall generation process. To address the above limitations, in this work, we introduce an automatic evaluation method that measures retrieval utility through the lens of information gain within the RAG framework. Specifically, we propose *Semantic Perplexity (SePer)*, a metric that captures the LLM's internal belief about the correctness of the generated answer. We then quantify the utility of retrieval by the extent to which it reduces semantic perplexity post retrieval. Extensive experiments demonstrate that SePer not only aligns closely with human preferences but also offers a more precise and efficient evaluation of retrieval utility across diverse RAG scenarios.

## 1 Introduction

Retrieval plays a crucial role in satisfying information needs across various interactive systems. With the rapid advancement of Large Language Models (LLMs) and various downstream applications Xu et al. (2025); Ye et al. (2025); Guo et al. (2025), retrieval has become deeply interwoven with generation processes Lewis et al. (2020); Guu et al. (2020). This integration not only enhances the accuracy and faithfulness of generated content Chen et al. (2017) but also enables handling more complex applications such as multi-hop reasoning Trivedi et al. (2022a), visual information seeking Hu et al. (2024), and task completion Yao et al. (2023); Zhang et al. (2023).

To evaluate and enhance these retrieval-augmented systems Salemi & Zamani (2024), a key challenge lies in measuring the contribution of retrieved information to the overall performance, i.e., the *utility of retrieval*. For instance, a reasoning process may require different pieces of information at different steps to infer the final answer Yang et al. (2018); Talmor & Berant (2018); Gu et al. (2024) as shown in Figure 1. However, most evaluation methods fail to respond to middle-step information, which may not directly match the ground truth text span. Besides, while a RAG workflow or agentic task might trigger retrieval multiple times within a single interaction cycle Asai et al. (2023); Jiang et al. (2023), it's difficult to quantify which retrieval effort brings in the most rewards. The lack of evaluator's sensitivity to partial information also results in discontinuous scoring of retrieved information Schaeffer et al. (2023), hampering the development of more efficient retrieval mechanisms.

Unlike the independent evaluation of retrievers, the utility of information retrieval (IR) hinges not only on the quality of the information but also on the prior knowledge of the recipient (e.g., LLM or human), their capacity to integrate external inputs and the way it interacts with the retriever Yoran et al. (2024); Shi et al. (2024). For example, a widely-known fact would bring no knowledge gain to LLMs, although it is both relevant and correct. In the meantime, an irrelevant long document may undermine LLM's performance even though it does not contain false facts and is harmless in general. Therefore, traditional IR metrics that evaluate the retriever as an independent module

---
*Corresponding authors.

cannot reflect its actual helpfulness from a systemic perspective. Recent works also use well-trained LLMs such as GPT-4 as generalizable judges to evaluate different aspects of RAG systems, such as relevancy, correctness, and faithfulness Es et al. (2024). However, these methods are often expensive and inefficient, especially for large-scale datasets and complex RAG workflows.

In this work, we propose the perspective to measure retrieval utility based on the knowledge gain of LLMs Belkin (1980); Belkin et al. (1982a;b). Ideally, an effective measure of information retrieval utility should reflect the satisfaction of the recipient Cooper (1973).

Historically, this approach has been hypothetical because the knowledge gain in real humans is intangible in computation. However, when LLMs act as information recipients, it is possible to estimate the shift in the LLM's knowledge distribution and use it as an indicator of retrieval effectiveness. Along this line, we define *Semantic Perplexity (SePer)*, a sampling-based method to estimate LLM's belief conditioned on an input query. Specifically, we first sample multiple responses and cluster them based on their semantic meanings and re-aggregate their likelihoods following the concept of semantic entropy Kuhn et al. (2023). In this way, we can compute probabilities in the semantic meaning space to obtain a more accurate

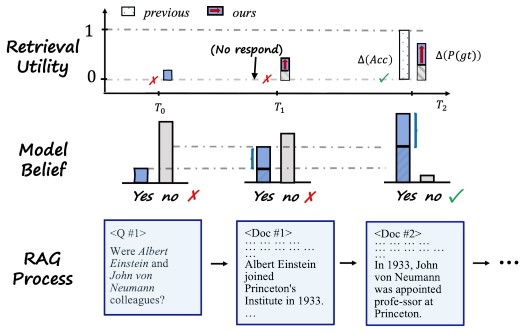

Figure 1: An illustration of retrieval utility in the multi-step RAG process. Unlike previous methods that only evaluate the final retrieval outcome, our approach can assess the utility of intermediate retrieval steps, even when the information retrieved is incomplete.

estimation of the belief distribution, as opposed to vocabulary space. We then compute the cross-entropy between the estimated semantic distribution and ground truth distribution, the exponential form of which is defined as *Semantic Perplexity (SePer)*. By doing so, we make it tangible to estimate the knowledge distribution shift of LLMs and use it to quantify retrieval utility.

In summary, our contributions are three-fold:

- We introduce $\Delta SePer$, an assessment method for evaluating retrieval utility based on shifts in LLMs' knowledge distributions. This approach not only aligns closer with human annotations but is also more consistent with inference time due to its sampling-based nature.
- We conduct theoretical analysis and extensive experiments to demonstrate that $\Delta SePer$ provides a more accurate, fine-grained, and efficient evaluation of retrieval utility. It is also generalizable across a broad range of RAG scenarios. Furthermore, we augment the evaluation of various RAG systems with our *SePer* metric for the reference of future research, which is maintained in `https://github.com/sepermetric/seper`.
- By utilizing *SePer*, we quantify the retrieval needs across different scenarios. Our findings offer valuable insights for data selection and budget allocation in practical RAG systems.

## 2 RELATED WORKS

**Evaluation of Information Retrieval**. The effectiveness of retrieval is often evaluated from two aspects: Direct content evaluation, which scrutinizes the relevance of the retrieved content itself, and response-based evaluation, which gauges the quality of the responses to reflect on the effectiveness of in-the-middle modules Salemi & Zamani (2024). However, each method suffers from several shortcomings individually. Direct context evaluation treats retrieval as an independent module, which cannot reflect the impact on the receiver model. Response-based methods can be further divided into two categories: model-based and reference-based methods. Model-based methods require knowledgable models, such as human and GPT-4 Liu et al. (2023), to evaluate whether the LLM output is a desired response to a given query. Reference-based methods require a reference answer and evaluate the LLM outputs by their matching score to the reference, such as BLEU Papineni et al. (2002), ROUGE Lin (2004), and BERTSCORE Zhang et al. (2020). However, these methods mainly reflect lexical information and cannot capture the nuanced relationship in semantic meaning. More recent works also leverage LLM judges, such as ARES Saad-Falcon et al. (2024), PROMETHEUS Kim et al. (2024), and AUTO-J Li et al. (2024) to assess the distance between reference and output. These LLM-based methods are often more accurate due to the language understanding and LLMs' comprehensive world knowledge. However, they are often slow and expensive.

**Knowledge Estimation in LLMs**. There is a view regarding LLM as a knowledge base Petroni et al. (2019); Geva et al. (2021), and generation can be viewed as retrieval from internal parametric memory Jiang et al. (2020). From this perspective, injecting external knowledge into the latent knowledge of LLM via in-context learning is equivalent to fine-tuning Dai et al. (2023). Xie et al. (2022) also explains in-context learning in a Bayesian inference framework and shows that prompts influence LLM output by shifting its latent concept distribution. While the internal knowledge latent distribution is unobservable, many studies have managed to estimate the knowledge state of LLMs from other signals, such as probing model internal states Ribeiro et al. (2016); Adi et al. (2017); Meng et al. (2022) and prompted responses Zhong et al. (2021). Studies also find that uncertainty in LLM is highly correlated with knowledge correctness, i.e., hallucination Kuhn et al. (2023); Cheng et al. (2024), which provides analysis on how knowledgeable LLM is by observing uncertainty.

## 3 QUANTIFY RETRIEVAL UTILITY

In this section, we justify our quantification of *retrieval utility* as a computable belief distribution shift in the information receiver model. This approach is grounded in the Bayesian framework, where new evidence updates prior beliefs. However, unlike traditional Bayesian updating that aims to learn model parameters, our focus is on evaluating the utility of the retrieved information in terms of its impact on the model's previous belief. We begin by envisioning several properties that *retrieval utility* should possess. We then formally define *retrieval utility* as the change of belief in ground-truth answers and prove that it satisfies the desirable properties. Finally, we instantiate this formulation in the RAG and introduce the algorithm details to compute *SePer* and retrieval utility.

### 3.1 NOTATIONS

Throughout this paper, we use the following notations. Let $M$ denote a well-trained language model (the *information receiver*) capable of generating answers to queries $q$. The correct answer set to the query $q$ is denoted by $\mathcal{A} = \{a^*\}$. The retrieved result is denoted by $\mathcal{D}$, where $\mathcal{D}$ is a set of $n$ atomic information $d_i$, i.e., $\mathcal{D}=\{d_1, d_2, ..., d_n\}$. We denote by $P_M(a)$ the likelihood model $M$ assigns to answer $a$ without retrieval, and by $P_M(a \mid \mathcal{D})$ the likelihood after incorporating $\mathcal{D}$.

### 3.2 RETRIEVAL UTILITY AS BELIEF REVISION

To provide an intuitive conceptualization of retrieval utility $U(M, d)$, we incorporate insights from cognitive information retrieval theories. These theories emphasize the dynamic interplay between information, the user's knowledge state, and the context of information retrieval. Consequently, we envision that an effective retrieval utility metric $U(M, d)$ should satisfy the following properties:

**Property 1.** *The retrieval utility $U(M, d)$ depends on both the retrieved information $d$ and the information receiver $M$.*

According to Cooper (1971; 1973); Dervin (1999); Ingwersen (1996), the effectiveness of a retrieval system is contingent upon both the user's existing knowledge and the relevance of the retrieved information. This perspective necessitates the introduction of dependence property, which considers both the information receiver and the retrieved content in evaluation.

**Property 2** (Zero Utility). *For a given query $q$, the retrieval utility $U(M, d)$ should be zero if the information $d$ retrieved is either irrelevant to $q$ or if the model $M$ already possesses the requisite knowledge to address $q$ effectively without $d$.*

Belkin et al. (1982a) posits that information is sought to resolve an anomaly in the user's knowledge state. Thus, if the retrieved information does not address this anomaly or if the user's knowledge is already sufficient, the information holds no utility, thereby justifying the zero utility property.

**Property 3** (Monotonicity). *Given an information receiver $M$ and an unsatisfied information need $q$, the retrieval utility $U(M, d)$ is a monotonically increasing function of the relevance of the retrieved information $d$ to $q$.*

According to Ingwersen (1996), $U(M, d)$ depends on information relevance and the user's cognitive space. With cognitive space fixed, increasing the relevance of retrieved information enhances utility, supporting the monotonicity property that $U(M, d)$ increases with the relevance of $d$ to $q$.

Intuitively, the retrieval utility quantifies how much the retrieved information $d$ shifts the model's belief toward the correct answer $a^*$. Accordingly, we define the retrieval utility as follows:

**Definition 1** (Retrieval Utility). *The retrieval utility is defined as the change in the model's belief about the correct answer $a^*$ due to the retrieved information $d$:*

$$U(M, d) = P_M(a^* \mid d) - P_M(a^*). \tag{1}$$

We demonstrate that Definition 1 satisfies the properties listed above.

*Proof of Property 1.* The retrieval utility $U(M, d)$ depends explicitly on both $d$ and $M$ through the probabilities $P_M(a^* \mid d)$ and $P_M(a^*)$.

*Proof of Property 2.* We discuss the two distinct scenarios in the property separately:

*1) Irrelevance of d to q:* When $d$ is irrelevant, it fails to contribute any new information relevant to the correct answer $a^*$. Consequently, the conditional probability of $a^*$ given $d$ equals the prior probability, $P_M(a^* \mid d) = P_M(a^*)$. Thus, the utility $U(M, d) = P_M(a^* \mid d) - P_M(a^*) = 0$.

*2) Redundancy of d for M:* If $M$ already knows $a^*$, the probability $P_M(a^*)$ is 1. Since probabilities cannot exceed 1, $P_M(a^* \mid d)$ also cannot exceed 1, implying $U(M, d) = P_M(a^* \mid d) - 1 = 0$. Here, since $d$ adds no value, $P_M(a^* \mid d) = P_M(a^*)$, and thus $U(M, d)$ remains 0.

*2) Redundancy of d for M:* Let $\mathbb{K}$ be the knowledge scope of model $M$. $a^*$ is known to $M$ means $P_M(a^*) = P_M(a^* \mid \mathbb{K}) = 1$. For redundant knowledge $d \in \mathbb{K}$, $P_M(a^* \mid \mathbb{K} \cup \{d\}) = P_M(a^* \mid \mathbb{K})$. Thus, $U(M, d) = P_M(a^* \mid d) - P_M(a^*) = 0$.

*Proof of Property 3.* Following Dai et al. (2024), let $d \vdash a^*$ denotes that $d$ entails $a^*$, we define the relevance of the retrieved information $d$ to the query $q$ as:

$$\text{Rel}(d, q) = \begin{cases} 1, & \text{if } d \vdash a^*, \\ 0, & \text{otherwise.} \end{cases} \tag{2}$$

When $\text{Rel}(d, q) = 0$, according to Property 2, the retrieval utility $U(M, d) = 0$. When $\text{Rel}(d, q) = 1$, since $d \vdash a^*$, assuming the receiver $M$ can effectively utilize $d$, we have $P_M(a^* \mid d) > P_M(a^*)$. Therefore, the retrieval utility is positive:

$$U(M, d) = P_M(a^* \mid d) - P_M(a^*) > 0. \tag{3}$$

Thus, as $\text{Rel}(d, q)$ increases from 0 to 1, $U(M, d)$ increases from 0 to a positive value, demonstrating the monotonicity property under this binary definition of relevance.

For more general cases where $\text{Rel}(d, q)$ is ordinal or continuous — for example, in multi-step reasoning where $d$ partially contributes to $a^*$ and $0 < \text{Rel}(d, q) < 1$ — we empirically demonstrate that our belief change based utility metric exhibits a significantly higher correlation with human-annotated context utility compared to other methods in Table 2.

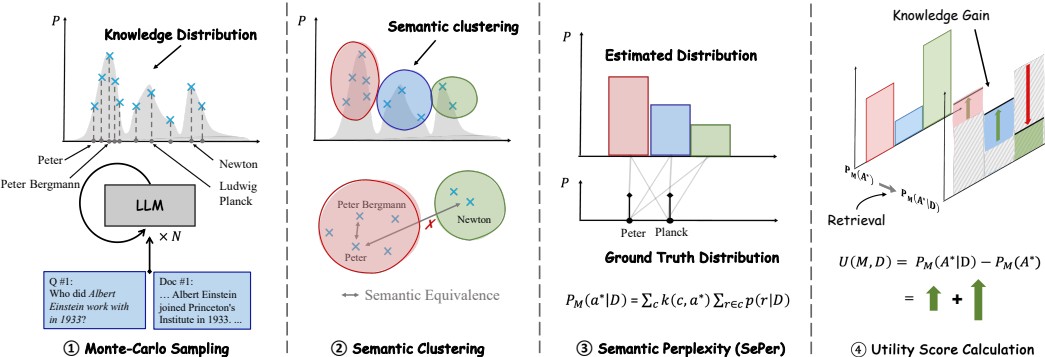

Figure 2: *SePer*: Estimating retrieval utility in multi-step retrieval-augmented generation (RAG) processes by measuring changes in model belief. *SePer* consists of four key steps: ① Probing the model's belief through Monte-Carlo Sampling, where the LM generates $N$ responses to the query using a temperature parameter. ② Estimating the belief distribution over possible answers using semantic clustering. ③ Calculating the model's semantic perplexity by comparing the estimated belief distribution with the ground truth distribution. ④ Assessing the unity of partial retrieval by measuring the change in semantic perplexity before and after retrieval.

### 3.3 BELIEF ESTIMATION THROUGH SAMPLING

Estimating model belief $P_M(a)$ is challenging due to the vast output space of the language model. Moreover, the model's outputs are in the vocabulary space, whereas our belief probabilities are defined over the semantic space. For example, "*Peter*", "*Peter Bergmann*" and "*Ludwig Planck*" are equally correct answers to the question "*Who did Einstein work with in 1933?*" and should be considered the same event in the probabilistic space Kuhn et al. (2023).

Extending semantic entropy Farquhar et al. (2024), we calculate the model's likelihood on $a^*$ based on the distribution of semantic meanings.

**Definition 2** (Semantic Equivalence). *Two texts $x$ and $y$ are semantically equivalent, denoted $x \equiv y$, if $x \vdash y$ and $y \vdash x$, where $\vdash$ means entailment.*

Practically, the entailment relationship is computed from a function $E : \mathcal{X} \times \mathcal{X} \to [0, 1]$, where $\mathcal{X}$ is the set of all possible texts. Given two texts $x$ and $y$, $E(x, y)$ outputs a score representing the degree to which $x$ entails $y$. The entailment relation holds if $E(x, y)$ exceeds a predefined threshold $\tau$. We also experimentally demonstrate that using the entailment model to define semantic equivalence Yao & Barbosa (2024) aligns much more accurately with human-annotated semantic equivalence, compared to traditional lexical-matching methods and trained LLM-judges, almost on par with GPT-4 evaluation. Results are shown in Table 1.

Given a set of responses $\{r_i\}$, *semantic clustering* is the process of grouping responses into clusters $\mathcal{C} = \{C_k\}$ such that all responses within a cluster are semantically equivalent:

$$C_k = \{r_i \mid r_i \equiv r_j, \forall r_j \in C_k\}. \tag{4}$$

The original semantic entropy makes the entropy computable in the sampled distribution:

$$\text{SE}(q) = -\sum_C \left( \left( \sum_{r \in C} p(r \mid q) \right) \log \left[ \sum_{r \in C} p(r \mid q) \right] \right) \approx \sum_{i=1}^{|\mathcal{C}|} -|C_i|^{-1} \log p(C_i \mid q). \tag{5}$$

Since the first equation is not computable due to infinite sentence space, the expectation is estimated using Monte-Carlo integration over sampled semantic cluster $\mathcal{C}$. By *the Law of Large Numbers*, as the number of samples $N \to \infty$, the frequency of responses converges to the model's belief distribution. We use a similar approximation in making *SePer* computable. But unlike semantic entropy, we estimate model belief shift on the reference answer instead of the output uncertainty originally for hallucination detection. We will detail the computation of *SePer* in the following part.

### 3.4 $\Delta$*SePer*: SEMANTIC PERPLEXITY REDUCTION IN RAG

To estimate model belief on reference answers $P(\{a^*\})$, instead of computing the entropy of semantic clusters, we further determine which clusters are semantically equivalent to any of the $a$ in $\{a^*\}$. For clarity, we begin with the special case where there is only one $a*$.

Thus, $P(a^*)$ is calculated by:

$$P(a^* \mid q) = \sum_C k(C, a^*) \sum_{r \in C} p(r \mid q), \tag{6}$$

where $r = \{t_1, t_2, \ldots, t_{i-1}\}$ and $p(r \mid q)$ is the output sequence likelihood from $M$:

$$p(r \mid q) = \prod_{i=1}^{|r|} p(t_i \mid t_1, t_2, \ldots, t_{i-1}, q). \tag{7}$$

$k(C, a^*)$ is a kernel function to measure the distance between the meaning of semantic cluster $C$ and $a^*$. We tested two different implementations of $k(C, a^*)$ the entailment model score $E(x, y)$ and Indicator function $I(C, a^*)$:

$$I(C, a^*) = \begin{cases} 1, & \text{if } C \equiv a^*, \\ 0, & \text{otherwise.} \end{cases} \tag{8}$$

Finally, the *SePer* score is calculated by:

$$\textit{SePer}_M(q, a^*) = P_M(a^* \mid q) \approx \sum_{C_i \in \mathcal{C}} k(C_i, a^*) p(C_i \mid q). \tag{9}$$

Similar to 5, the right approximate equation is an unbiased estimator of the left one. This naturally extends to the general cases where there are multiple ground-truth $a^*$ provided:

$$\textit{SePer}_M(q, \mathcal{A}) \approx \frac{1}{|\mathcal{A}|} \sum_{a^* \in \mathcal{A}} \left[ \sum_{C_i \in \mathcal{C}} k(C, a^*) p(C_i \mid q) \right]. \tag{10}$$

---

**Algorithm 1** *SePer* & $\Delta$*SePer*

---

**Require:** Model $M$, reference answer $a^*$, entailment model $E$, threshold $\tau$, number of samples $N$.
1: Sample responses: $r_i \sim P_M(r)$, for $i = 1, \ldots, N$.
2: Compute likelihoods: $\ell_i = P_M(r_i)$.
3: *SePer$_H$:*                                                      *SePer$_S$:*

    **1.** Semantic clustering: Group responses $r_j$ into clusters     **1.** Compute entailment scores:
        $\mathcal{C} = \{C_k\}$ such that $E(r_i, r_j) \geq \tau$ within each cluster.         $k_i = E(r_i, a^*)$.
    **2.** Identify $C_{a^*}$: matching reference answer with semantic     **2.** Compute soft belief:
        cluster, where $\forall r \in C_{a^*}, r \equiv a^*$.         $P_M(a^*) = \sum_{i=1}^{N} \ell_i \cdot k_i$.
    **3.** Compute semantic perplexity: $P_M(a^*) = \sum_{r_i \in C_{a^*}} \ell_i$.
4: Repeat steps with retrieved information $\mathcal{D}$ to obtain $P_M(a^* \mid \mathcal{D})$.
5: Compute utility: $U(M, \mathcal{D}) = \Delta$*SePer* $= P_M(a^* \mid \mathcal{D}) - P_M(a^*)$.

---

Lastly, retrieval utility $U(M, \mathcal{D})$ is calculated by $\Delta$*SePer*, i.e., semantic perplexity reduction:

$$U(M, \mathcal{D}) = \Delta\textit{SePer} = P_M(a^* \mid q, \mathcal{D}) - P_M(a^* \mid q). \tag{11}$$

Through Monte-Carlo sampling and semantic clustering, $\Delta$*SePer* quantifies the extent to which receiver $M$'s belief shifts towards ground-truth answer after retrieval, i.e., how much beneficial information gain the information pieces brought to the model. Based on two different kernel function choices, we implemented *SePer$_S$* and *SePer$_H$* separately. The incorporation of kernel-based soft matching provides a more nuanced and continuous evaluation Nikitin et al. (2024). The *SePer* and $\Delta$*SePer* algorithms are fully described in Algorithm 1.

## 4 EVALUATION OF *SePer*

In this section, we conducted experiments to prove the validity and reliability of the proposed *SePer* metric Xiao et al. (2023); Jacobs & Wallach (2021); Wagner et al. (2021). For validity testing, we first show experiments in Section 4.1.1 to prove that *SePer* is a more accurate and fine-grained indicator for reference-based response evaluation and then demonstrate its better correlation and alignments with human judgments about retrieval utility in Section 4.1.2. For reliability testing, we test the robustness of the performance of *SePer* on different aspects, including varying datasets, repeated computation, and the number of samples used. Results in Section 4.2 show that under our default setting, *SePer* achieved high reliability and stability with less cost in time and money. We also add ablation results about more hyperparameters in A to prove the robustness of *SePer*.

### 4.1 VALIDITY OF *SePer*

#### 4.1.1 VALIDITY OF SEMANTIC-BASED ANSWER SCORING

First, we evaluate the basic component of computing *SePer*, i.e., using the entailment model to calculate the semantic similarity between the generated answer and the ground-truth answer.

**Datasets.** We use EVOUNA Wang et al. (2024), a Question Answering (QA) benchmark to evaluate QA evaluators' reliability. Based on NATURAL QUESTIONS (NQ) Kwiatkowski et al. (2019) and TRIVIAQA Joshi et al. (2017), EVOUNA augmented the datasets with LLM-generated responses and asked humans to annotate whether a response is semantically equivalent to the golden answer.

**Baselines.** We include two types of baselines: Matching-based evaluation, such as lexical match and BERTSCORE, and LLM judge evaluators, such as AUTO-J and PROMETHEUS.

**Model.** We use the `deberta-v2-xlarge-mnli`[1] He et al. (2021) model fine-tuned on the MNLI dataset to assess the entailment relationship between two text pairs following the setting of Kuhn et al. (2023), which is far more efficient than API-based entailment judgment Yao & Barbosa (2024) without a significant performance drop. In our implementation, we further leverage the entailment score to get a probabilistic estimation of the likelihood of semantic equivalence.

**Results.** As shown in Table 1, the NLI-based evaluation in *SePer* demonstrates significantly higher alignment with human judgment compared to traditional matching-based response evaluation by surpassing the baselines by 2% $\sim$ 6% F1-score across various generators and datasets. Notably, it is close to or minorly surpasses the response evaluation performance of GPT-4 in this benchmark. BERTSCORE, while capturing semantic meaning, may fail to capture the relationships in QA tasks. At the same time, trained LLM judges did not demonstrate an edge over traditional methods on this

---

[1] `https://huggingface.co/microsoft/deberta-v2-xlarge-mnli`

benchmark, with reference-free judges falling far behind reference-based methods. These results show that computing the semantic-equivalence score based on the entailment model is both an efficient and reliable method. Its high correlation score in matching responses and answers also set a solid step for the next stage of computation of $\Delta SePer$.

| Evaluator | Natural Questions | | | | | TriviaQA | | | | |
|---|---|---|---|---|---|---|---|---|---|---|
| | DPR-FiD F1/Acc | InstructGPT F1/Acc | ChatGPT F1/Acc | GPT-4 F1/Acc | BingChat F1/Acc | DPR-FiD F1/Acc | InstructGPT F1/Acc | ChatGPT F1/Acc | GPT-4 F1/Acc | BingChat F1/Acc |
| **Matching-based** | | | | | | | | | | |
| Lexical Match | 92.0/89.7 | 86.9/84.8 | 85.0/80.3 | 87.6/82.5 | 87.8/82.3 | 91.8/94.7 | 94.8/92.3 | 95.2/92.3 | 94.8/91.1 | 94.1/89.8 |
| BERTScore | 83.5/75.1 | 77.6/69.5 | 81.2/72.8 | 84.3/76.0 | 77.5/67.5 | 75.1/65.5 | 84.1/75.7 | 88.4/80.8 | 90.5/93.5 | 88.3/80.4 |
| Entail (*SePer*) | 96.6/95.3 | 92.0/90.1 | 91.2/87.8 | 93.1/89.7 | 91.4/87.0 | 97.6/96.1 | 97.5/96.0 | 97.9/96.4 | 98.5/97.2 | 96.2/93.2 |
| **LLM-as-a-Judge** | | | | | | | | | | |
| Auto-J | 57.8/54.2 | 71.9/62.1 | 76.4/66.5 | 75.4/65.1 | 72.8/62.2 | 76.3/66.7 | 80.8/71.5 | 81.4/71.3 | 80.4/68.7 | 83.0/73.0 |
| PROMETHEUS | 83.8/77.8 | 81.1/70.5 | 86.4/77.7 | 89.3/81.5 | 89.5/82.3 | 89.4/83.1 | 90.0/83.2 | 93.0/87.7 | 94.7/90.2 | 95.4/91.8 |
| **Human-level** | | | | | | | | | | |
| GPT-4 | 96.0/94.5 | 93.2/91.0 | 93.7/90.6 | 95.1/92.0 | 94.7/91.4 | 98.3/97.3 | 98.4/97.5 | 98.5/97.5 | 98.8/97.8 | 98.1/96.5 |
| Human | 97.4/96.3 | 97.8/96.8 | 96.5/95.6 | 97.9/96.6 | 97.2/95.5 | 100/100 | 99.6/99.4 | 99.2/98.8 | 99.2/99.8 | 99.9/99.8 |

Table 1: Correlation of entailment-based answer scoring (*SePer*) with human answer scoring. We use the F1-score and Accuracy to measure the degree of correlation. As shown in the table, *SePer* achieves the highest accuracy in answer scoring and is on par with human-level judgments.

### 4.1.2 VALIDITY OF $\Delta SePer$ ON QUANTIFYING RETRIEVAL UTILITY

Secondly, we prove that using $\Delta SePer$ in measuring retrieval utility is highly correlated with human annotations with a larger margin than baseline methods. We test our method in two different settings: 1) Simple question-answering tasks, which generally require a single document for answer generation, and 2) reasoning-involved question answering, which requires collecting and integrating several steps of partial information to correctly solve a problem.

**Datasets.** In the simple open QA setting, we use three representative datasets: NQ, MS MARCO Bajaj et al. (2016), and SQUAD Rajpurkar et al. (2016). Each of these datasets has annotations of questions, golden answers, and human-annotated positive/negative passages. Since answering the question requires only one passage, we first attach the positive passages with a utility score of 1 and the negative passages with a utility score of 0. Since positive passages can not bring utility to LLMs based on their known knowledge, we then filter out those cases in which LLM has already succeeded in each dataset to eliminate the baseline effect. Through this preprocessing, we got the utility label on passages from real-human annotations. In the reasoning-involved QA setting, we use four typical Multihop-QA datasets, 2WIKIMULTIHOPQA Ho et al. (2020), HOT-POTQA Yang et al. (2018), IIRC Ferguson et al. (2020), and MUSIQUE Trivedi et al. (2022b), which contains annotations of positive passages in the middle steps of a reasoning chain. Since each step only contains incomplete information pieces, we make a natural assumption that the overall information utility is uniformly assigned to each step and thus attach a ground-truth utility score of $1/n_{\text{steps}}$ for each middle-step passage. While not perfect, we find this assumption reasonable since these datasets are mostly collected by means of question composition, as detailed in Appendix I.

**Metrics.** We use the Pearson correlation score to measure the correlation between our $\Delta SePer$ score and ground-truth utility score. We use $t$-test to assess the significance of the observed correlation coefficient, with statistic $t$ computed with $t = r \times \sqrt{(n-2)/(1-r^2)}$. We then map the $t$ to $p$-value using the Student's $t$-distribution table. As a result, all the Pearson correlation coefficients in Table 2 have corresponding $p$-values less than 0.01, providing strong evidence against the null hypothesis and indicating a high level of statistical significance.

**Baselines.** We choose various methods that can be used to estimate the LLM's knowledge and use their difference to measure retrieval utility. Lexical-matching-based methods include EM, ROUGE, and BLEU, which measure the response correctness score through matching text spans. BERTSCORE matches predicted answers and ground truth through embedding similarity. Another category of baselines is uncertainty measures, such as perplexity, entropy, and semantic entropy. Unlike the matching-based method, these uncertainty measures do not require golden answers in computation. While they are not directly defined on the correctness dimension, we include them as recent literature also shows that uncertainty is correlated with knowledge capabilities Farquhar et al. (2024); Cheng et al. (2024) in calibrated LLMs. We also included LLM-judges similar to Table 1.

**Implementation details.** We use the same semantic equivalence scoring algorithm as tested in Table 1. For each query, we sample $k = 10$ times and obtain the response along with sequence likelihood. All baseline methods are sampled for the same $k$, and the final score comes from mean aggregation. In Table 2, we use `Llama-2-7b-chat-hf`[2] as the generator LLM. We also tested other sizes and showed a tendency results in Figure 4.

**Results.** We show the result in Table 2. $\Delta SePer$ scores show significant improvement on other metrics. Specifically, $\Delta SePer_S$ has a marginal improvement on $\Delta SePer_H$ across different datasets, which may indicate that soft probability mass assignment can capture more nuanced meanings, especially in free-form generations. Comparing simple and reasoning QA tasks, the scores of almost all different metrics are lower by $\sim 10\%$, indicating the challenging nature of reasoning-based QA. Even though, $\Delta SePer$ can achieve a Pearson correlation score greater than 0.5 across almost all datasets, showing its great potential to act as an automatic evaluation metric for retrieval utility.

| Method | Simple | | | Reasoning | | | |
|---|---|---|---|---|---|---|---|
| | NQ | MS MARCO | SQUAD | 2WIKIMHQA | HOTPOTQA | IIRC | MUSIQUE |
| Exact Match | 0.454 | 0.197 | 0.422 | 0.307 | 0.392 | 0.303 | 0.298 |
| ROUGE | 0.691 | 0.443 | 0.808 | 0.482 | 0.578 | 0.399 | 0.489 |
| BLEU | 0.188 | 0.353 | 0.298 | 0.197 | 0.206 | 0.126 | 0.163 |
| BERTSCORE | 0.592 | 0.322 | 0.564 | 0.361 | 0.451 | 0.197 | 0.392 |
| PERPLEXITY | 0.008 | 0.005 | 0.024 | 0.005 | 0.013 | 0.009 | 0.011 |
| Entropy | 0.431 | 0.142 | 0.557 | 0.226 | 0.276 | 0.292 | 0.203 |
| Semantic Entropy | 0.491 | 0.171 | 0.621 | 0.262 | 0.339 | 0.342 | 0.258 |
| AUTO-J | 0.421 | 0.022 | 0.406 | 0.243 | 0.183 | 0.096 | 0.169 |
| PROMETHEUS | 0.639 | 0.307 | 0.707 | 0.502 | 0.508 | 0.383 | 0.464 |
| $\Delta SePer_H$ | 0.752 | 0.512 | 0.904 | 0.559 | 0.634 | 0.446 | 0.543 |
| $\Delta SePer_S$ | **0.769** | **0.533** | **0.905** | **0.584** | **0.660** | **0.461** | **0.559** |

Table 2: Pearson correlation between different evaluation metrics and ground-truth retrieval utility with $p$-value $< 0.01$ for $\Delta SePer$. As shown in the table, both $\Delta SePer_H$ and $\Delta SePer_S$ significantly outperform other baselines in measuring retrieval utility in simple and reasoning-type tasks, with the soft version leading an edge.

## 4.2 RELIABILITY OF *SePer*

We further test the reliability of *SePer* from different aspects, i.e., whether *SePer* produces consistent and stable evaluation across different datasets, random repetitions, and number of samples.

We tested $\Delta SePer_H$ and $\Delta SePer_S$ with different numbers of samples across four datasets: NQ, HOTPOTQA, MS MARCO, and SQUAD, and the results are shown in Figure 3. We choose the number of sampled responses $n$ at $\{1, 5, 10, 15, 20\}$ for ablation purposes, extending the default choice of $n = 10$. We find that the conclusion is consistent in different datasets: As $n$ increases, the correlation of $\Delta SePer$ with ground truth also increases, indicating better accuracy. Besides, the variance generally becomes smaller as n increases, indicating better robustness. Generally, the elbow point appears at $n = 5$ to $n = 10$, with $n = 10$ having less than a

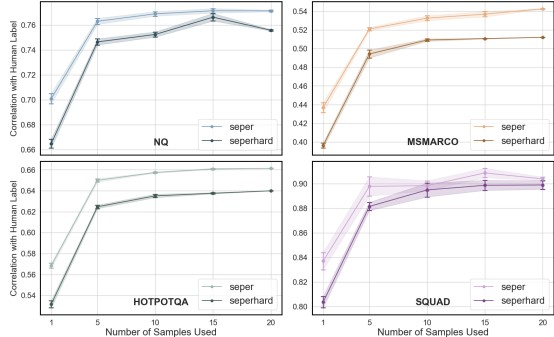

Figure 3: Influence of the number of samples and repeated calculation of *SePer* on four datasets.

1% performance drop compared to $n = 20$. Thus, using $n = 10$ in $\Delta SePer$ computation would be an effective choice.

---

[2] https://huggingface.co/meta-llama/Llama-2-7b-chat-hf

The shadow area and error bar in Fig.3 shows that the fluctuation of *SePer*'s quality among repeated calculations is less than 1%, indicating high stability according to measurement theory. More experiments and ablation about the robustness of *SePer* can be found in Appendix A.

## 5 FINDINGS BASED ON *SePer*

In this section, we apply *SePer* to different modules in the RAG pipeline and exhibit our findings through the new lens of *SePer*. In general, RAG pipelines use techniques such as reranker, refiner, and control flow to improve generation quality. Through the unique lens of *SePer*, we can get a more fine-grained and accurate view of how these factors affect the overall performance of RAG. We benchmark current RAG workflows in Appendix C. A brief introduction of these components in RAG and experiment details can be found in Appendix D.

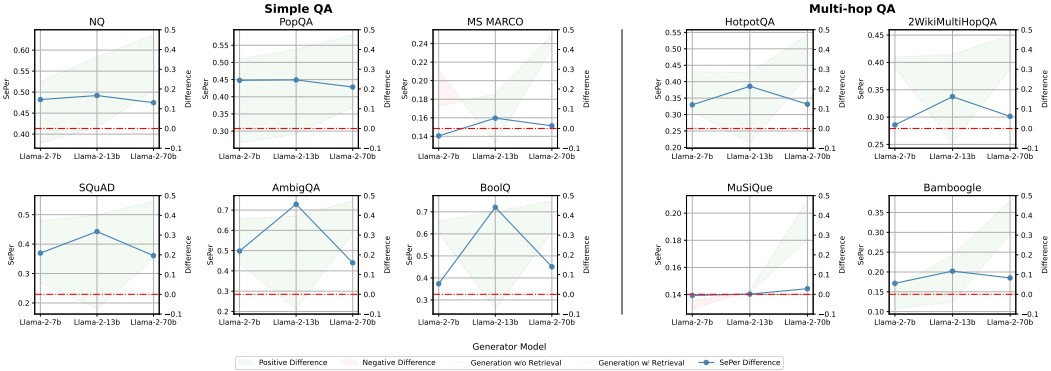

Figure 4: Results about applying *SePer* on different RAG settings. The ▨ and ▨ areas represent the positive and negative differences between *SePer* for generation w/ and w/o retrieval, respectively. The solid blue line indicates $\Delta SePer$, i.e., the utility of retrieval. The red dashed line indicates the zero point of the differences.

### 5.1 EXPERIMENT GOALS

We aim to address the following main research questions (RQs) through the lens of *SePer*. RQ1-4 aims to observe the utility of RAG components on the final performance, including retrieval, reranker, and prompt compression. Specifically, RQ1 and RQ2 look closer at the retrieval utility and what impact on RAG can be brought by varying numbers of retrieved items and the choice of generator models of different sizes. These RQs are all designed to provide evidence and guidance on designing more efficient and effective RAG pipelines:

- **RQ1:** What is the utility of retrieval on LLMs of different sizes?
- **RQ2:** How does the number of retrieved items influence overall RAG performance?
- **RQ3:** How do different prompt compression methods influence the overall RAG performance?
- **RQ4:** How does the reranking phase influence the overall RAG performance?

### 5.2 RETRIEVAL UTILITY FOR GENERATOR MODELS OF VARYING SIZES (RQ1)

In figure 4, we evaluate how LLMs of different sizes can benefit from retrieval. Our experiments are conducted on both simple QA and multi-hop QA datasets, and more implementation details can be found in Appendix D. We observed that 1) for both scenarios of QA tasks, models of different sizes generally make positive use of retrieved information to produce better answers for most datasets. An exception is MS MARCO, which we attribute to its source corpus being different from the Wikipedia corpus we uniformly used. 2) According to our experiments, medium-size models benefit the most from retrieved information. This could be due to 1) their weaker initial knowledge without retrieval and 2) their better ability to absorb retrieved in-context information as compared to smaller models.

### 5.3 UTILITY OF DIFFERENT NUMBERS OF RETRIEVED ITEMS (RQ2)

The number of in-context retrieved items used in prompts, noted as $k$, can significantly impact the model's generation results. Figure 5(a) shows the experiments about the impact of $k$ on the overall RAG performance, with $k$ set at $\{1, 5, 10\}$. For most datasets, retrieving more information progressively positively affects answering questions. However, increasing the number of in-context

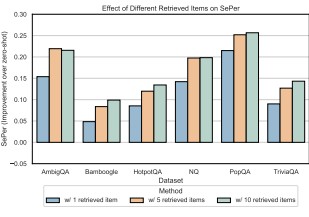 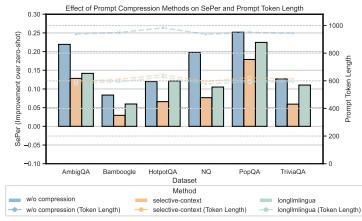 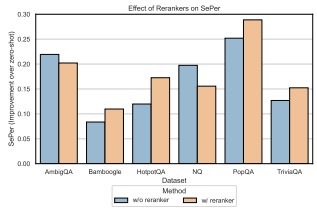

Figure 5: Differences in *SePer* across various retrieval and generation settings. Panel (a) illustrates the differences in *SePer* between generations w/ and w/o retrieval, analyzed under different retrieved items. Panel (b) highlights the effect of prompt compression methods on *SePer* differences compared to generation w/o retrieval. Panel (c) examines the impact of the reranker on *SePer* differences relative to generation w/o retrieval.

items from 5 to 10 only brings marginal improvement and sometimes even slightly hurts the generation performance, as in the AmbigQA dataset. This might be due to the extra noise brought by an increasing number of retrieved documents.

### 5.4 UTILITY OF PROMPT COMPRESSION METHODS (RQ3)

The prompt compression module is used to reduce the prompt length to lower the inference cost while preserving or facilitating the RAG performance.

Figure 5(b) illustrates the results of our experiments. We test the utility of two major prompt compression works, SELECTIVE-CONTEXT Li et al. (2023) and LONGLLMLINGUA Jiang et al. (2024). Although both prompt compression methods slightly reduce *SePer* compared to no compression, both methods can reduce the prompt by about 40%, thus lowering the inference costs to about one-third of the original. Additionally, we note that the LONGLLMLINGUA maintains a relatively higher *SePer* than selective-context, becoming a preferred choice for balancing performance and inference cost. More details about prompt compression methods can be found in Appendix D.1.

### 5.5 UTILITY OF RERANKER (RQ4)

While retrieval can quickly gather candidate items from large document collections to aid generation, it often lacks precision in small $k$, which leaves out important information and brings in many noises. To this end, the reranker module is introduced to the RAG pipeline, which not only selects relevant documents into prompt contexts with better accuracy but also re-arranges them in the best order for overall generation quality Liu et al. (2024). More details about the lines of work on reranker can be found in Appendix D.2.

Figure 5(c) shows experimental results comparing $\Delta SePer$ with and without rerankers from the implementation of `bge-reranker-large`[3] Xiao et al. (2024). We set top-$k$ values of 20 for retrieved items and 5 for reranked items in reranked scenarios while keeping a constant top-$k$ of 5 in non-reranked scenarios. Results from $\Delta SePer$ are consistent with the conclusions of works in the field that reranker, in general, brings significant improvement to the RAG pipeline by removing noises and reordering contexts. However, in NQ and AMBIGQA (which are also derived from NQ), it seems that the reranking process has a negative impact on answer quality. This might indicate that simply putting more relevant contexts at an earlier position may not be the best strategy. How the ordering of contexts influences the final generation results is open for exploration.

## 6 DISCUSSION

This study introduces Semantic Perplexity (*SePer*) and then $\Delta SePer$, a novel metric that evaluates the utility of information retrieval by measuring the knowledge gain in large language models (LLMs). *SePer* provides a more nuanced understanding of retrieval effectiveness beyond mere accuracy, aligning closer with real-world inference needs.

Our findings demonstrate that $\Delta SePer$ can effectively quantify retrieval needs across various scenarios, aiding in data selection and resource allocation in RAG systems. This metric can enhance the optimization of RAG systems for both efficiency and effectiveness, promising improved performance in complex AI applications. Future work will focus on extending *SePer*'s applicability to more diverse and challenging RAG scenarios.

---

[3] https://huggingface.co/BAAI/bge-reranker-large

## 7 ACKNOWLEDGMENT

This work was supported in part by the National Key R&D Program of China (Grant No. 2023YFF0725001), in part by the National Natural Science Foundation of China (Grant No.92370204), in part by the Guangdong Basic and Applied Basic Research Foundation (Grant No. 2023B1515120057), in part by Guangzhou-HKUST(GZ) Joint Funding Program (Grant No. 2023A03J0008), Education Bureau of Guangzhou Municipality. Yijie Xu acknowledges the support from the modern matter laboratory, HKUST(GZ).

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

# Part I

# Appendix

## Table of Contents

# A    DETAILS OF EXPERIMENTS

## A.1    DATASET STATISTICS

This section presents the datasets used in our experiments, detailing their names, sizes, sources, key characteristics, and availability, which are categorized by task type (Single QA, Multi-hop QA, Fact Verification, Multiple-choice QA, and Summarization). The table below also includes links to the original papers and dataset repositories.

| Dataset | Size | Source | Key Characteristics | Availability |
|---|---|---|---|---|
| **Single QA** | | | | |
| **MS MARCO** Bajaj et al. (2016) | 101,093 | Bing | Search engine queries, web passages | [Paper] [Dataset] |
| **POPQA** Mallen et al. (2023) | 14,267 | Wikipedia | Focuses on popular culture questions | [Paper] [Dataset] |
| **TRIVIAQA** Joshi et al. (2017) | 11,313 | Wikipedia & Web | Trivia questions with evidence documents | [Paper] [Dataset] |
| **SQUAD** Rajpurkar et al. (2016) | 10,570 | Wikipedia | Standard reading comprehension | [Paper] [Dataset] |
| **NQ** Kwiatkowski et al. (2019) | 3,610 | Wikipedia | Real user queries, diverse topics | [Paper] [Dataset] |
| **BOOLQ** Clark et al. (2019) | 3,270 | Wikipedia | Yes/No questions, requires inference | [Paper] [Dataset] |
| **AMBIGQA** Min et al. (2020) | 2,002 | Wikipedia | Question ambiguity with multiple answers | [Paper] [Dataset] |
| **FERMI** Kalyan et al. (2021) | 1,000 | Wikipedia | Estimation-based reasoning | [Paper] [Dataset] |
| **Multi-hop QA** | | | | |
| **HOTPOTQA** Yang et al. (2018) | 7,405 | Wikipedia | Requires multi-hop reasoning | [Paper] [Dataset] |
| **2WIKIMULTIHOPQA** Ho et al. (2020) | 12,576 | Wikipedia | Cross-document multi-hop reasoning | [Paper] [Dataset] |
| **MUSIQUE** Trivedi et al. (2022b) | 2,417 | Wikipedia | Multi-hop reasoning, complex questions | [Paper] [Dataset] |
| **BAMBOOGLE** Press et al. (2023) | 125 | Wikipedia | Compositional reasoning challenging for Google | [Paper] [Dataset] |
| **Fact Verification** | | | | |
| **FEVER** Thorne et al. (2018) | 10,444 | Wikipedia | Fact verification | [Paper] [Dataset] |
| **Multiple-choice QA** | | | | |
| **MMLU** Hendrycks et al. (2021) | 14,042 | N/A | Multiple-choice, general knowledge | [Paper] [Dataset] |
| **Summarization** | | | | |
| **WIKIASP** Hayashi et al. (2021) | 37,368 | Wikipedia | Open-domain summarization | [Paper] [Dataset] |

Table 3: Summary of Datasets Used in Our Experiments

## A.2    ROBUSTNESS OF *SePer* IN REPEATED TESTS

| Dataset | # Repetition | # Samples | $\sigma$ | | Coefficient Variance | |
|---|---|---|---|---|---|---|
| | | | *SePer* | **Correlation** | *SePer* | **Correlation** |
| **NQ** Kwiatkowski et al. (2019) | 5 | 10 | 0.053 | 0.002 | 0.028 | 0.002 |
| **MS MARCO** Bajaj et al. (2016) | 5 | 10 | 0.055 | 0.003 | 0.003 | 0.005 |
| **HOTPOTQA** Yang et al. (2018) | 5 | 10 | 0.045 | 0.001 | 0.037 | 0.001 |
| **SQUAD** Rajpurkar et al. (2016) | 5 | 10 | 0.056 | 0.004 | 0.063 | 0.004 |

Table 4: Extended experimental details on the robustness test of *SePer*. As shown in the table, *SePer* demonstrates stability upon repeated testing in our default choice of ten samples.

Table 4 shows that the standard deviation($\sigma$) and coefficient variation in calculating *SePer* is less than 10%, which means that *SePer* produces the result of low variance in repeated tests. We also calculated the variation in the degree of correlation with human judgments in repeated tests. Results show that the fluctuation of the correlation score is less than 1%. All these experimental signs prove that the proposed *SePer* is a reliable and stable measurement according to the theory of statistics.

### A.2.1    $P$-VALUES WITH THE PEARSON CORRELATION OF $\Delta$*SePer*

| Dataset | NQ | MS MARCO | SQUAD | HOTPOTQA | 2WIKIMHQA | MUSIQUE | IIRC |
|---|---|---|---|---|---|---|---|
| $p$-value | 0 | 0 | $2.86 \times 10^{-207}$ | 0 | 0 | 0 | $2.88 \times 10^{-218}$ |

Table 5: $p$-values across different datasets. The $p$-value 0 does not necessarily mean zero, but below the numerical precision, which indicates a number very close to 0.

# B EXTENDED RESULTS

## B.1 QUALITATIVE ANALYSIS

In this section, we analyze two RAG cases qualitatively to demonstrate the effectiveness of the proposed $\Delta SePer$ in measuring retrieval utility.

We use two cases from two typical scenarios: one for simple RAG, in which the answer is contained in a single document, and another for reasoning-intensive RAG, in which the answer should be reasoned over multiple documents.

| Question: | Who sings does he love me with reba? |
|---|---|
| **Reference Docs:** | **Doc1**: "Does He Love You" is a song written by Sandy Knox and Billy Stritch, and recorded as a duet by American country music artists Reba McEntire and Linda Davis. It was released in August 1993 as the first single from Rebaś album "Greatest Hits Volume Two". It is one of country musicś several songs about a love triangle. "Does He Love You" was written in 1982 by Billy Stritch. He recorded it with a trio in which he performed at the time, because he wanted a song that could be sung by the other two members. |
| **Ground Truth Answer** | Linda Davis. |

| Retrieved Docs Doc1 | Model Answer (x10) | GT Answer | $\Delta SePer$ | Amount of information |
|---|---|---|---|---|
| ✗ | Reba McEntire: 10 | Linda Davis | 0 | 0 |
| ✓ | Linda Davis: 10 | | 1.0 | 1 |

Table 6: Case #1 of simple RAG task: $\Delta SePer$ on single retrieved doc. $\Delta SePer$ accurately reflects the utility of retrieved documents.

**Results Analysis of Case #1:** Results in Table 6 shows that when no useful document is provided (the ✗ means the retrieved document is irrelevant), the model consistently fails to answer the question correctly, even with ten times' sampling. At this time, the calculated $\Delta SePer$ is 0, accurately indicating the zero utility of irrelevant information. When a positive document is retrieved, the model successfully generates the correct answer. At this time, the calculated $\Delta SePer$ is 1, accurately indicating the utility of useful information.

| Question: | Are the Laleli Mosque and Esma Sultan Mansion located in the same neighborhood? |
|---|---|
| **Reference Docs:** | **Doc1**: The Laleli Mosque (Turkish: 'Laleli Camii', or Tulip Mosque) is an 18th-century Ottoman imperial mosque located in Laleli, Fatih, Istanbul, Turkey. **Doc2**: The Esma Sultan Mansion (Turkish: 'Esma Sultan Yalısı'), a historical yalı (English: waterside mansion) located on the Bosphorus in the Ortaköy neighborhood of Istanbul, Turkey, named after its original owner, Esma Sultan, is now used as a cultural center after redevelopment. |
| **Ground Truth Answer** | No. |

| Retrieved Docs | | Model Answer (x10) | GT Answer | $\Delta SePer$ | Amount of information |
|---|---|---|---|---|---|
| Doc1 | Doc2 | | | | |
| ✗ | ✗ | Yes: 10, No: 0 | | 0 | 0 |
| ✗ | ✓ | Yes: 7, No: 3 | No | 0.15 | 1/2 |
| ✓ | ✗ | Yes: 8, No: 2 | | 0.1 | 1/2 |
| ✓ | ✓ | Yes: 3, No: 7 | | 0.7 | 1 |

Table 7: Case #2 of reasoning-based RAG task: $\Delta SePer$ on multiple retrieved docs. $\Delta SePer$ reflects the utility of retrieved information in a fine-grained way, successfully responding to partial information.

**Results Analysis of Case #2:** Results in Table 7 demonstrate that when no relevant retrieval is provided, the model consistently gives the incorrect answer, "Yes," which indicates it is unable to infer the relationship between the two locations. When only one piece of contextual information is made available, the model's answers begin to vary. This outcome suggests that partial information, although not sufficient for producing the correct answer consistently, causes the model to reconsider its initially confident but incorrect response. Traditional evaluation methods often assess retrieval utility based solely on whether the retrieved information directly enables the model to provide a correct answer and overlook the intermediate benefits that partial information can offer. In contrast, our $\Delta SePer$ successfully responds to even partial information, providing a fine-grained evaluation of information utility.

## B.2 EFFICIENCY ANALYSIS

We conduct a latency and cost analysis using widely available commercial LLM APIs. Specifically, we evaluate multiple APIs with varying pricing structures from providers, including OpenAI, Anthropic, Google, and Deepseek.

In the *Direct Evaluation* setting, we prompt the LLM with a given question, context, and answer and request the model to assess the contribution of the context to the overall response by assigning an integer score between 1 and 10. While in the *Reduction Evaluation* setting, we first query the LLM with the combination of query, context, and answer to evaluate the correctness of the answer. Subsequently, we query the LLM with only the query and answer to assess the correctness without the context. The difference between these two scores is computed to determine the $\Delta SePer$.

For our experiments, we utilize the Natural Questions Kwiatkowski et al. (2019) dataset, selecting 10 questions with corresponding references collected from passages in the Wikipedia corpus using the E5 Wang et al. (2022) model. We report the average prompt length and the average time consumed across the ten questions. We list the user prompts used in the experiment as follows:

---

**Prompt for *Direct Evaluation* in LLM APIs**

Evaluate the contribution of the given context to the provided answer for the specified question.
Your evaluation should be based on how effectively the context supports or justifies the answer.
Provide your assessment using an integer rating between 1 (minimal or no contribution) and 10 (critical or complete contribution).
Do not output any other information or context.
- Question: {question}
- Context: {context}
- Answer: {answer}
Your evaluation:

---

**Prompt for *Reduction Evaluation* in LLM APIs with context**

Evaluate the correctness of the given answer based on the question and the provided context.
Rate correctness using an integer between 1 (completely incorrect) and 10 (completely correct).
Only provide the rating as your output.
- Question: {question}
- Context: {context}
- Answer: {answer}
Your rating:

---

---

**Prompt for *Reduction Evaluation* in LLM APIs without context**

Evaluate the correctness of the given answer based solely on the question.
Ignore any external information and rate correctness using an integer between 1 (completely incorrect) and 10 (completely correct).
Only provide the rating as your output.
- Question: {question}
- Answer: {answer}
Your rating:

---

Table 8 summarizes the average time latencies and costs across different API providers. We checked the latest pricing on the official websites of each API and did not enable any potential batching mechanisms. Additionally, it demonstrates the advantages of our proposed *SePer* and $\Delta$*SePer* in terms of time and economic costs.

| Models | Company | Direct Evaluation | | Reduction Evaluation | |
|---|---|---|---|---|---|
| | | Time (s) | Cost ($) | Time (s) | Cost ($) |
| CHATGPT-4O-LATEST | OpenAI[1] | 4.22 | 0.0077 | 6.13 | 0.0080 |
| GPT-4-TURBO | OpenAI[1] | 2.01 | 0.0155 | 3.80 | 0.0163 |
| GPT-3.5-TURBO | OpenAI[1] | 3.34 | 0.0008 | 3.89 | 0.0008 |
| CLAUDE-3-5-SONNET-NX | Anthropic[2] | 7.45 | 0.0046 | 30.30 | 0.0052 |
| CLAUDE-3-HAIKU-NX | Anthropic[2] | 5.63 | 0.0003 | 19.21 | 0.0004 |
| GEMINI-1.5-PRO | Google[3] | 3.29 | 0.0192 | 6.97 | 0.0203 |
| GEMINI-1.5-FLASH | Google[3] | 3.56 | 0.0001 | 6.26 | 0.0001 |
| DEEPSEEK-CHAT | Deepseek[4] | 0.88 | 0.0000 | 1.68 | 0.0000 |
| *SePer & $\Delta$SePer* | N/A | **0.12** | **Free** | **0.24** | **Free** |

Table 8: **Latency ($\downarrow$) and Cost ($\downarrow$) Comparison of Various LLM Models in Direct and Reduction Evaluation Settings.** The table shows the average response time (in seconds) and cost (in USD) for each model in Direct and Reduction evaluation tasks.

### B.3 CIRCUMSTANCES OF NEGATIVE UTILITY

Additionally, our experiments, conducted according to the settings outlined in Figure 6, revealed that some retrieved items negatively impact question-answering performance. We extended our tests to additional datasets to further investigate this phenomenon, and the datasets involved are listed in Table 3.

We first observed that in certain datasets, the retrieved items hindered the model's question-answering ability. For instance, in the MMLU dataset, which is a multiple-choice dataset with relatively straightforward questions, the model can often rely on its own knowledge to answer correctly. In such cases, the retrieved items proved detrimental. For the MS MARCO dataset, we attributed the performance issue to distribution differences, as the corpus source (Bing) differs from the source of other datasets (Wikipedia). For more complex datasets like MUSIQUE, 2WIKIMUL-TIHOPQA, and FERMI, which require multi-step reasoning and logical chains, a small number of retrieved items could not provide all the necessary information. However, when enough items were retrieved, they offered a more comprehensive information set, thereby assisting the model in making correct inferences. Additionally, In the FEVER dataset, which focused on Fact Verification, an excessive number of retrieved items disrupted the model's ability to verify facts effectively.

Regarding prompt compression methods, excluding datasets that already showed negative effects even without compression (as discussed in the previous paragraph), the FERMI dataset—rich in

---

[1] https://openai.com/api/pricing/
[2] https://www.anthropic.com/pricing#anthropic-api
[3] https://ai.google.dev/pricing
[4] https://api-docs.deepseek.com/quick_start/pricing

mathematical logits—was particularly affected by incorrect token compression, resulting in errors. Similarly, in the 2WIKIMULTIHOPQA dataset, the compression of logical chains was identified as a major issue.

Despite these challenges, we still found that the use of the reranker consistently improved performance across these datasets, further validating its robustness.

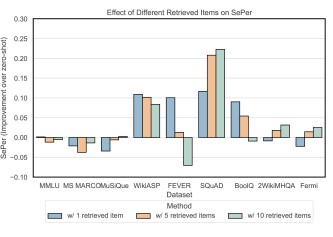 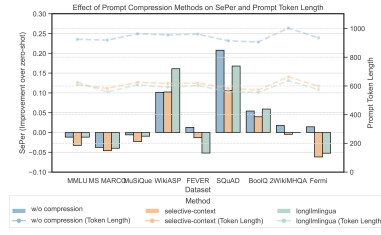 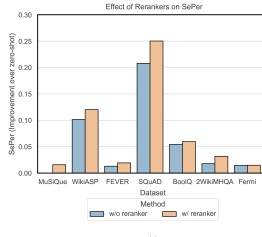

Figure 6: Differences in *SePer* under various retrieval and generation settings. Same as Figure 5, Panel (a) shows the differences in *SePer* for generation with and without retrieval under different numbers of retrieved information. Panel (b) illustrates the impact of using prompt compression and which compression method on the differences in *SePer* compared to generation without retrieval. Panel (c) demonstrates the reranker's effect compared to generation without retrieval.

### B.4 ABLATION OF THE CONSISTENCY OF *SePer* WITH DIFFERENT ENTAILMENT MODELS

In this paper, we choose `deberta-v2-xlarge-mnli`[4] He et al. (2021) following Farquhar et al. (2024) for deciding the entailment relationship, which strikes a balance between accuracy and efficiency. For the sake of the ablation study, we tested the influence of the choice of entailment models on the result of *SePer* in Figure 7. We choose seven mainstream models used for NLI classification tasks with varying sizes and architectures and compute the Pearson correlation of *SePer* produced from these models. Specifically, the entailment models we used are listed as follows:

| Model | Developer | Size (Parameters) | # of Layers | Hidden Size | Architecture |
|---|---|---|---|---|---|
| DeBERTa-Base-MNLI[1] | Microsoft | 86M | 12 | 768 | Encoder-only |
| DeBERTa-Large-MNLI[2] | Microsoft | 350M | 24 | 1024 | Encoder-only |
| DeBERTa-XLarge-MNLI[3] | Microsoft | 700M | 48 | 1024 | Encoder-only |
| DeBERTa-V2-XLarge-MNLI[4] | Microsoft | 710M | 24 | 1536 | Encoder-only |
| DeBERTa-V2-XXLarge-MNLI[5] | Microsoft | 1.3B | 48 | 1536 | Encoder-only |
| RoBERTa-Large-MNLI[6] | Facebook | 355M | 24 | 1024 | Encoder-only |
| BART-Large-MNLI[7] | Facebook | 406M | 12+12 | 1024 | Encoder-Decoder |

Table 9: Details of different entailment models for ablation study on the robustness of *SePer*. We choose main-stream models used in the field of NLI, covering different sizes and architectures.

We tested 7 NLI models on 9 datasets and 3 different choices for $k$ (number of items retrieved and used in in-context prompting) in computing *SePer*.

According to the results in Figure 7, all of the Pearson correlation scores between the results of model pairs are above 0.7. Specifically, for simple-type QA tasks (NQ, AMBIGQA, MSMARCO, SQUAD, TRIVIAQA, POPQA), the entailment judgments are even more consistent, with all scores above 0.85 and most scores above 0.9. For reasoning-type QA tasks (HotpotQA, 2WikiMultihopQA, MuSiQue), the entailment scores are all above 0.7, with most scores above 0.9.

---

[1] https://huggingface.co/microsoft/deberta-base-mnli
[2] https://huggingface.co/microsoft/deberta-large-mnli
[3] https://huggingface.co/microsoft/deberta-xlarge-mnli
[4] https://huggingface.co/microsoft/deberta-v2-xlarge-mnli
[5] https://huggingface.co/microsoft/deberta-v2-xxlarge-mnli
[6] https://huggingface.co/FacebookAI/roberta-large-mnli
[7] https://huggingface.co/facebook/bart-large-mnli

Based on these high correlation scores among different entailment models, we can safely draw the conclusion that *SePer* is robust in computation in terms of entailment model choice, and it is still effective and reliable on small models (86M) when high efficiency is required.

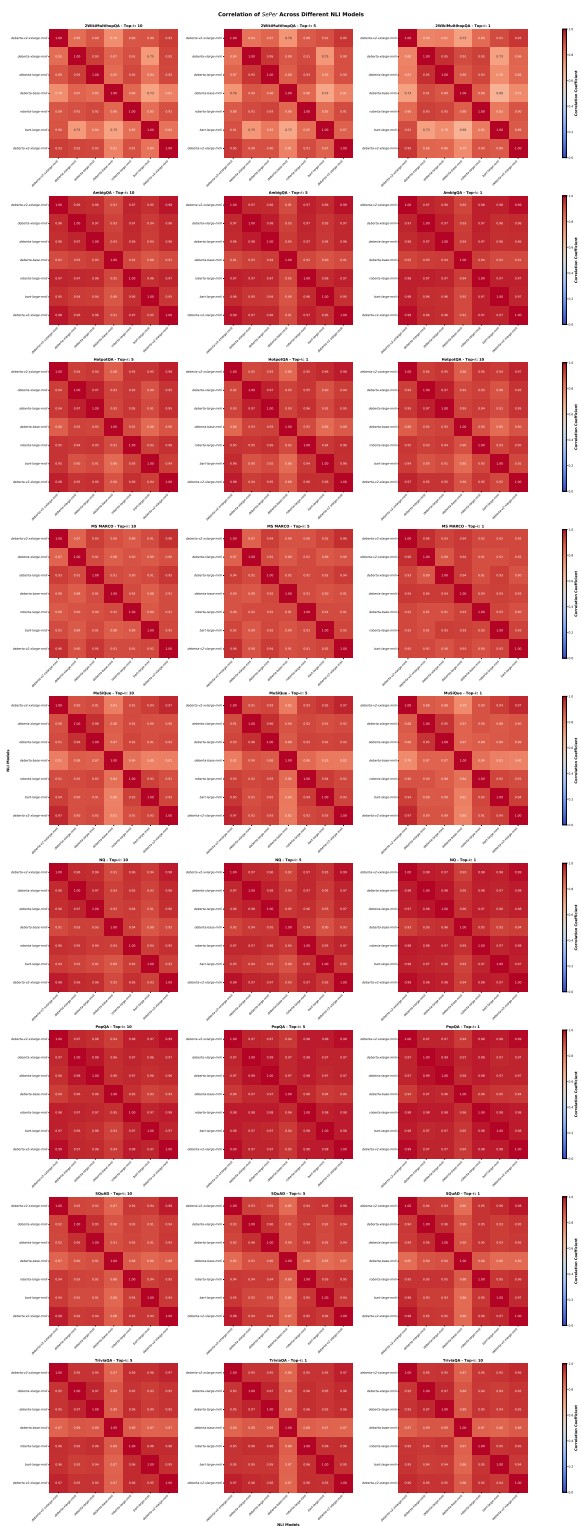

Figure 7: Consistency of *SePer* computation among different entailment models.

## C  BENCHMARKING RETRIEVER AND WORKFLOW

For more detailed information on benchmarking the retriever and the entire workflow, please visit the following link: *SePer* Benchmarks.

### C.1  BENCHMARKING RETRIEVER QUALITY AND UTILITY

The majority of dense retrievers in dense retrieval paradigms are based on transformer architectures such as BERT. This opens the possibility for further fine-tuning with more extensive and higher-quality datasets, as well as more advanced algorithms. Consequently, a wide variety of retrievers have emerged in the community. To evaluate their performance, we present benchmarks based on multiple datasets (described in Table 3) and multiple retrievers, utilizing *SePer* as the evaluation framework. These benchmarks aim to assess both the quality and utility of different retrievers.

For quality evaluation, we provide a benchmark comparing the performance of retrievers on standard retrieval tasks with various numbers of retrieved items. For utility evaluation, we propose a $\Delta$*SePer*-based benchmark. In this setup, the $\Delta$*SePer* is computed by taking the difference between the *SePer* scores achieved using *question-answer* pairs and those obtained using *question-retrieval context-answer* pairs.

We evaluate six dense retrievers: $\text{AAR}_{\text{ANCE}}$ Yu et al. (2023), $\text{AAR}_{\text{Contriever}}$ Yu et al. (2023), BGE Xiao et al. (2024), Contriever Izacard et al. (2022), DPR Karpukhin et al. (2020), and E5 Wang et al. (2022), alongside the classical sparse retriever BM25 Robertson et al. (1995). The results of the quality and utility benchmarks are presented in Tables 10 and 11, respectively.

### C.2  BENCHMARKING WORKFLOW UTILITY

Naive RAG strictly follows the *retrieval-generation* paradigm, which limits its ability to utilize retrieved information for further retrieval. This limitation is critical for complex reasoning tasks, such as multi-hop question answering. Therefore, recent research has proposed several workflows that enable the entire RAG pipeline to perform multiple retrievals and integrate information, which may enhance the reasoning ability of large language models.

We benchmarked four RAG workflows—RETROBUST Yoran et al. (2024), FLARE Jiang et al. (2023), IRCoT Trivedi et al. (2022a), and ITER-RETGEN Shao et al. (2023)—on multiple datasets using varying numbers of retrieved items. Since these methods may involve multiple rounds of retrieval, existing retrieval metrics, such as retrieval recall, are no longer suitable. Thus, we only report the $\Delta$*SePer* metric. Additionally, RETROBUST only provides LoRA checkpoints for Llama 2 Touvron et al. (2023) 13B, so results for the 7B model are marked as "N/A." The results of the benchmark are shown in Table 12.

| Metric | | SePer | | | | | | | | | | Retrieval Recall | | |
|---|---|---|---|---|---|---|---|---|---|---|---|---|---|---|
| Top-k | | 1 | | | 5 | | | 10 | | | | 1 | 5 | 10 |
| Dataset name | Retriever | 7B | 13B | 70B | 7B | 13B | 70B | 7B | 13B | 70B | | | | N/A |
| **NQ** | AAR-ANCE | 0.323 | 0.321 | 0.383 | 0.413 | 0.424 | 0.493 | 0.446 | 0.461 | 0.533 | | 0.331 | 0.574 | 0.647 |
| | AAR-contriever | 0.385 | 0.376 | 0.445 | 0.481 | 0.496 | 0.563 | 0.497 | 0.516 | 0.593 | | 0.391 | 0.670 | 0.754 |
| | bge | 0.464 | 0.449 | 0.517 | 0.530 | 0.537 | 0.598 | 0.535 | 0.551 | 0.609 | | 0.507 | 0.740 | 0.800 |
| | bm25 | 0.282 | 0.270 | 0.347 | 0.367 | 0.377 | 0.457 | 0.409 | 0.431 | 0.505 | | 0.248 | 0.463 | 0.562 |
| | contriever | 0.301 | 0.309 | 0.368 | 0.409 | 0.421 | 0.501 | 0.448 | 0.472 | 0.544 | | 0.272 | 0.539 | 0.632 |
| | dpr | 0.403 | 0.389 | 0.448 | 0.469 | 0.482 | 0.560 | 0.490 | 0.510 | 0.586 | | 0.424 | 0.662 | 0.722 |
| | e5 | 0.504 | 0.489 | 0.555 | 0.557 | 0.563 | 0.620 | 0.560 | 0.565 | 0.621 | | 0.570 | 0.774 | 0.833 |
| **TRIVIAQA** | AAR-ANCE | 0.550 | 0.577 | 0.640 | 0.605 | 0.635 | 0.708 | 0.626 | 0.671 | 0.708 | | 0.444 | 0.632 | 0.698 |
| | AAR-contriever | 0.627 | 0.629 | 0.689 | 0.684 | 0.699 | 0.761 | 0.702 | 0.723 | 0.761 | | 0.548 | 0.725 | 0.776 |
| | bge | 0.629 | 0.624 | 0.684 | 0.675 | 0.690 | 0.750 | 0.697 | 0.711 | 0.750 | | 0.580 | 0.738 | 0.795 |
| | bm25 | 0.606 | 0.607 | 0.684 | 0.632 | 0.662 | 0.729 | 0.659 | 0.688 | 0.729 | | 0.508 | 0.688 | 0.739 |
| | contriever | 0.554 | 0.579 | 0.656 | 0.634 | 0.656 | 0.720 | 0.659 | 0.688 | 0.720 | | 0.427 | 0.659 | 0.734 |
| | dpr | 0.587 | 0.607 | 0.667 | 0.667 | 0.688 | 0.745 | 0.682 | 0.709 | 0.745 | | 0.555 | 0.740 | 0.786 |
| | e5 | 0.677 | 0.667 | 0.736 | 0.714 | 0.727 | 0.783 | 0.730 | 0.739 | 0.783 | | 0.635 | 0.779 | 0.818 |
| **2WIKIMULTIHOPQA** | AAR-ANCE | 0.348 | 0.341 | 0.344 | 0.371 | 0.358 | 0.404 | 0.380 | 0.403 | 0.429 | | 0.175 | 0.300 | 0.348 |
| | AAR-contriever | 0.357 | 0.349 | 0.348 | 0.383 | 0.375 | 0.412 | 0.398 | 0.414 | 0.452 | | 0.189 | 0.336 | 0.419 |
| | bge | 0.368 | 0.350 | 0.371 | 0.403 | 0.391 | 0.431 | 0.408 | 0.420 | 0.453 | | 0.216 | 0.405 | 0.467 |
| | bm25 | 0.374 | 0.367 | 0.378 | 0.392 | 0.398 | 0.428 | 0.403 | 0.434 | 0.460 | | 0.244 | 0.390 | 0.457 |
| | contriever | 0.353 | 0.349 | 0.353 | 0.373 | 0.380 | 0.403 | 0.385 | 0.407 | 0.435 | | 0.154 | 0.273 | 0.335 |
| | dpr | 0.326 | 0.322 | 0.309 | 0.352 | 0.341 | 0.370 | 0.362 | 0.368 | 0.399 | | 0.150 | 0.244 | 0.299 |
| | e5 | 0.366 | 0.354 | 0.361 | 0.389 | 0.393 | 0.427 | 0.402 | 0.424 | 0.465 | | 0.223 | 0.384 | 0.467 |
| **HOTPOTQA** | AAR-ANCE | 0.297 | 0.295 | 0.365 | 0.339 | 0.345 | 0.444 | 0.347 | 0.364 | 0.474 | | 0.205 | 0.367 | 0.428 |
| | AAR-contriever | 0.376 | 0.351 | 0.445 | 0.402 | 0.394 | 0.507 | 0.409 | 0.409 | 0.519 | | 0.287 | 0.465 | 0.538 |
| | bge | 0.392 | 0.378 | 0.464 | 0.423 | 0.423 | 0.537 | 0.436 | 0.448 | 0.543 | | 0.326 | 0.573 | 0.639 |
| | bm25 | 0.383 | 0.361 | 0.468 | 0.422 | 0.427 | 0.535 | 0.430 | 0.440 | 0.535 | | 0.311 | 0.496 | 0.564 |
| | contriever | 0.309 | 0.305 | 0.380 | 0.352 | 0.341 | 0.457 | 0.369 | 0.376 | 0.479 | | 0.198 | 0.359 | 0.440 |
| | dpr | 0.314 | 0.301 | 0.360 | 0.335 | 0.340 | 0.440 | 0.350 | 0.361 | 0.454 | | 0.229 | 0.376 | 0.438 |
| | e5 | 0.378 | 0.373 | 0.460 | 0.421 | 0.414 | 0.527 | 0.428 | 0.442 | 0.545 | | 0.307 | 0.534 | 0.610 |
| **PopQA** | AAR-ANCE | 0.395 | 0.394 | 0.415 | 0.467 | 0.480 | 0.513 | 0.478 | 0.490 | 0.551 | | 0.431 | 0.662 | 0.726 |
| | AAR-contriever | 0.375 | 0.372 | 0.394 | 0.418 | 0.435 | 0.477 | 0.431 | 0.452 | 0.504 | | 0.403 | 0.579 | 0.645 |
| | bge | 0.456 | 0.454 | 0.474 | 0.489 | 0.513 | 0.550 | 0.477 | 0.515 | 0.566 | | 0.510 | 0.704 | 0.766 |
| | bm25 | 0.287 | 0.282 | 0.301 | 0.334 | 0.339 | 0.379 | 0.358 | 0.366 | 0.427 | | 0.286 | 0.437 | 0.488 |
| | contriever | 0.297 | 0.291 | 0.325 | 0.356 | 0.367 | 0.408 | 0.382 | 0.386 | 0.442 | | 0.281 | 0.446 | 0.508 |
| | dpr | 0.330 | 0.321 | 0.344 | 0.396 | 0.409 | 0.445 | 0.397 | 0.425 | 0.465 | | 0.355 | 0.532 | 0.593 |
| | e5 | 0.474 | 0.466 | 0.486 | 0.512 | 0.528 | 0.571 | 0.512 | 0.535 | 0.590 | | 0.528 | 0.729 | 0.790 |

Table 10: *SePer* Benchmark Across Retrievers and Parameter Sizes. The colors highlight the best-performing retrievers under each dataset.

| Dataset name | Retriever | ΔSePer Top-1 7B | 13B | 70B | ΔSePer Top-5 7B | 13B | 70B | ΔSePer Top-10 7B | 13B | 70B | Retrieval Recall 1 | 5 (N/A) | 10 |
|---|---|---|---|---|---|---|---|---|---|---|---|---|---|
| NQ | AAR-ANCE | -0.033 | -0.068 | -0.101 | 0.057 | 0.036 | 0.009 | 0.090 | 0.072 | 0.049 | 0.331 | 0.574 | 0.647 |
| | AAR-contriever | 0.029 | -0.012 | -0.039 | 0.125 | 0.108 | 0.079 | 0.141 | 0.127 | 0.109 | 0.391 | 0.670 | 0.754 |
| | bge | 0.108 | 0.060 | 0.033 | 0.174 | 0.149 | 0.114 | 0.178 | 0.162 | 0.125 | 0.507 | 0.740 | 0.800 |
| | bm25 | -0.074 | -0.118 | -0.137 | 0.010 | -0.012 | -0.027 | 0.053 | 0.042 | 0.021 | 0.248 | 0.463 | 0.562 |
| | contriever | -0.055 | -0.079 | -0.116 | 0.053 | 0.033 | 0.017 | 0.091 | 0.083 | 0.060 | 0.272 | 0.539 | 0.632 |
| | dpr | 0.047 | 0.000 | -0.036 | 0.113 | 0.093 | 0.076 | 0.134 | 0.121 | 0.102 | 0.424 | 0.662 | 0.722 |
| | e5 | 0.148 | 0.100 | 0.071 | 0.201 | 0.174 | 0.136 | 0.204 | 0.176 | 0.137 | 0.570 | 0.774 | 0.833 |
| TRIVIAQA | AAR-ANCE | -0.031 | -0.059 | -0.116 | 0.025 | -0.001 | -0.048 | 0.017 | 0.035 | -0.025 | 0.444 | 0.632 | 0.698 |
| | AAR-contriever | 0.046 | -0.007 | -0.067 | 0.104 | 0.063 | 0.006 | 0.035 | 0.087 | 0.019 | 0.548 | 0.725 | 0.776 |
| | bge | 0.049 | -0.012 | -0.071 | 0.095 | 0.054 | -0.005 | 0.045 | 0.075 | 0.015 | 0.580 | 0.738 | 0.795 |
| | bm25 | 0.025 | -0.029 | -0.072 | 0.052 | 0.025 | -0.027 | 0.040 | 0.051 | -0.005 | 0.508 | 0.688 | 0.739 |
| | contriever | -0.026 | -0.057 | -0.099 | 0.053 | 0.020 | -0.036 | 0.022 | 0.051 | -0.008 | 0.427 | 0.659 | 0.734 |
| | dpr | 0.007 | -0.029 | -0.088 | 0.086 | 0.052 | -0.011 | -0.001 | 0.073 | 0.010 | 0.555 | 0.740 | 0.786 |
| | e5 | 0.097 | 0.031 | -0.020 | 0.133 | 0.091 | 0.028 | 0.039 | 0.103 | 0.034 | 0.635 | 0.779 | 0.818 |
| 2WIKIMULTIHOPQA | AAR-ANCE | -0.015 | 0.008 | -0.021 | 0.008 | 0.025 | 0.039 | 0.017 | 0.070 | 0.063 | 0.175 | 0.300 | 0.348 |
| | AAR-contriever | -0.006 | 0.015 | -0.017 | 0.020 | 0.042 | 0.047 | 0.035 | 0.081 | 0.087 | 0.189 | 0.336 | 0.419 |
| | bge | 0.005 | 0.016 | 0.006 | 0.040 | 0.058 | 0.065 | 0.045 | 0.087 | 0.088 | 0.216 | 0.405 | 0.467 |
| | bm25 | 0.011 | 0.034 | 0.013 | 0.030 | 0.065 | 0.062 | 0.040 | 0.101 | 0.095 | 0.244 | 0.390 | 0.457 |
| | contriever | -0.009 | 0.016 | -0.013 | 0.010 | 0.047 | 0.038 | 0.022 | 0.074 | 0.070 | 0.154 | 0.273 | 0.335 |
| | dpr | -0.037 | -0.011 | -0.056 | -0.011 | 0.008 | 0.005 | -0.001 | 0.035 | 0.033 | 0.150 | 0.244 | 0.299 |
| | e5 | 0.003 | 0.021 | -0.004 | 0.027 | 0.060 | 0.062 | 0.039 | 0.091 | 0.100 | 0.223 | 0.384 | 0.467 |
| HOTPOTQA | AAR-ANCE | 0.003 | -0.002 | -0.039 | 0.045 | 0.048 | 0.040 | 0.053 | 0.067 | 0.070 | 0.205 | 0.367 | 0.428 |
| | AAR-contriever | 0.083 | 0.054 | 0.041 | 0.108 | 0.097 | 0.103 | 0.115 | 0.112 | 0.115 | 0.287 | 0.465 | 0.538 |
| | bge | 0.098 | 0.081 | 0.060 | 0.129 | 0.126 | 0.132 | 0.142 | 0.151 | 0.138 | 0.326 | 0.573 | 0.639 |
| | bm25 | 0.089 | 0.064 | 0.064 | 0.129 | 0.130 | 0.130 | 0.136 | 0.143 | 0.131 | 0.311 | 0.496 | 0.564 |
| | contriever | 0.015 | 0.009 | -0.024 | 0.058 | 0.044 | 0.053 | 0.075 | 0.079 | 0.075 | 0.198 | 0.359 | 0.440 |
| | dpr | 0.020 | 0.004 | -0.044 | 0.041 | 0.043 | 0.035 | 0.056 | 0.064 | 0.050 | 0.229 | 0.376 | 0.438 |
| | e5 | 0.084 | 0.076 | 0.056 | 0.127 | 0.117 | 0.122 | 0.134 | 0.145 | 0.140 | 0.307 | 0.534 | 0.610 |
| POPQA | AAR-ANCE | 0.124 | 0.110 | 0.058 | 0.196 | 0.196 | 0.156 | 0.207 | 0.206 | 0.194 | 0.431 | 0.662 | 0.726 |
| | AAR-contriever | 0.104 | 0.088 | 0.037 | 0.147 | 0.151 | 0.119 | 0.160 | 0.168 | 0.147 | 0.403 | 0.579 | 0.645 |
| | bge | 0.185 | 0.170 | 0.117 | 0.218 | 0.229 | 0.193 | 0.206 | 0.231 | 0.209 | 0.510 | 0.704 | 0.766 |
| | bm25 | 0.016 | -0.002 | -0.057 | 0.063 | 0.055 | 0.022 | 0.087 | 0.082 | 0.069 | 0.286 | 0.437 | 0.488 |
| | contriever | 0.026 | 0.007 | -0.032 | 0.086 | 0.083 | 0.051 | 0.111 | 0.102 | 0.085 | 0.281 | 0.446 | 0.508 |
| | dpr | 0.059 | 0.037 | -0.013 | 0.125 | 0.125 | 0.087 | 0.127 | 0.141 | 0.108 | 0.355 | 0.532 | 0.593 |
| | e5 | 0.203 | 0.182 | 0.128 | 0.241 | 0.244 | 0.214 | 0.241 | 0.251 | 0.233 | 0.528 | 0.729 | 0.790 |

Table 11: ΔSePer Benchmark Across Retrievers and Parameter Sizes. The colors highlight the best-performing retrievers under each dataset.

| Metric | | \triangle*SePer* | | | | | |
|---|---|---|---|---|---|---|---|
| Top-$k$ | | 1 | | 5 | | 10 | |
| Dataset name | Workflow | 7B | 13B | 7B | 13B | 7B | 13B |
| **2WIKIMULTIHOPQA** | NAIVE | 0.366 | 0.354 | 0.389 | 0.393 | 0.402 | 0.424 |
| | RETROBUST | N/A | 0.623 | N/A | 0.644 | N/A | 0.700 |
| | FLARE | 0.360 | 0.358 | 0.369 | 0.336 | 0.374 | 0.344 |
| | IRCOT | 0.339 | 0.365 | 0.364 | 0.388 | 0.386 | 0.405 |
| | ITER-RETGEN | 0.371 | 0.346 | 0.420 | 0.413 | 0.435 | 0.446 |
| **HOTPOTQA** | NAIVE | 0.378 | 0.373 | 0.421 | 0.414 | 0.428 | 0.442 |
| | RETROBUST | N/A | 0.537 | N/A | 0.575 | N/A | 0.589 |
| | FLARE | 0.283 | 0.286 | 0.290 | 0.271 | 0.299 | 0.277 |
| | IRCOT | 0.362 | 0.418 | 0.409 | 0.452 | 0.442 | 0.466 |
| | ITER-RETGEN | 0.396 | 0.380 | 0.447 | 0.447 | 0.465 | 0.486 |
| **MUSIQUE** | NAIVE | 0.087 | 0.089 | 0.116 | 0.125 | 0.128 | 0.137 |
| | RETROBUST | N/A | 0.456 | N/A | 0.464 | N/A | 0.485 |
| | FLARE | 0.110 | 0.140 | 0.113 | 0.133 | 0.115 | 0.135 |
| | IRCOT | 0.143 | 0.131 | 0.161 | 0.157 | 0.176 | 0.164 |
| | ITER-RETGEN | 0.109 | 0.106 | 0.143 | 0.159 | 0.156 | 0.178 |
| **NQ** | NAIVE | 0.504 | 0.489 | 0.557 | 0.563 | 0.560 | 0.565 |
| | RETROBUST | N/A | 0.580 | N/A | 0.605 | N/A | 0.594 |
| | FLARE | 0.333 | 0.234 | 0.343 | 0.222 | 0.340 | 0.235 |
| | IRCOT | 0.457 | 0.500 | 0.493 | 0.534 | 0.510 | 0.515 |
| | ITER-RETGEN | 0.497 | 0.476 | 0.540 | 0.547 | 0.561 | 0.559 |
| **POPQA** | NAIVE | 0.474 | 0.466 | 0.512 | 0.528 | 0.512 | 0.535 |
| | RETROBUST | N/A | 0.493 | N/A | 0.553 | N/A | 0.521 |
| | FLARE | 0.328 | 0.248 | 0.347 | 0.244 | 0.343 | 0.247 |
| | IRCOT | 0.426 | 0.451 | 0.478 | 0.496 | 0.483 | 0.493 |
| | ITER-RETGEN | 0.462 | 0.447 | 0.499 | 0.512 | 0.486 | 0.523 |
| **TRIVIAQA** | NAIVE | 0.677 | 0.667 | 0.714 | 0.727 | 0.730 | 0.739 |
| | RETROBUST | N/A | 0.778 | N/A | 0.815 | N/A | 0.811 |
| | FLARE | 0.555 | 0.504 | 0.558 | 0.486 | 0.568 | 0.490 |
| | IRCOT | 0.597 | 0.688 | 0.671 | 0.722 | 0.704 | 0.730 |
| | ITER-RETGEN | 0.688 | 0.677 | 0.730 | 0.734 | 0.740 | 0.751 |

Table 12: $\triangle$*SePer* Benchmark Across Workflow and Parameter Sizes. The colors highlight the best-performing retrievers under each dataset.

## D    EXPERIMENT DETAILS IN SECTION 5

To demonstrate that our proposed *SePer* and $\Delta$*SePer* effectively integrate with various RAG pipelines, we conduct extensive experiments in Section 5. We also aim to show that *SePer* and $\Delta$*SePer* are module-agnostic within RAG pipelines.

Following the taxonomy proposed by Jin et al. (2024); Gao et al. (2023), modern modular RAG systems consist of various interchangeable and combinable modules, including refiner and reranker. These modules can be adapted or replaced to better target specific downstream tasks, providing greater flexibility and task-specific optimization. We additionally selected the **prompt compression** (a kind of refiner) and **reranker** modules for benchmarking and aim to provide a detailed explanation of their mechanisms and roles here.

### D.1    PROMPT COMPRESSION

Prompt compression shortens the prompt by filtering redundant and low-value content while ensuring the context fits within the model's context length.

Given a large language model (LLM) and an input prompt $x$, let the response generated by the model be denoted as $\text{LLM}(x)$. The goal of prompt compression is to find a compressed prompt $x'$ such that:

$$\mathcal{D}\big(\mathcal{P}_{\text{LLM}(x)}, \mathcal{P}_{\text{LLM}(x')}\big) < \epsilon, \tag{12}$$

where $\mathcal{P}_{\text{LLM}(x)}$ and $\mathcal{P}_{\text{LLM}(x')}$ represent the distributions over the model's responses when prompted with $x$ and $x'$, respectively. These distributions reflect the stochasticity introduced by sampling methods (e.g., temperature scaling, top-$k$, or nucleus sampling) during text generation.

Here, $\mathcal{D}(\cdot, \cdot)$ denotes a divergence metric, such as KL divergence, computed in a semantically meaningful space. Since the responses are text, we embed them in a suitable representation space (e.g., sentence embeddings) where these metrics can effectively measure differences in meaning and style.

The compression requirement is formalized as:

$$\text{len}(x') < \text{len}(x). \tag{13}$$

This ensures that $x'$ retains the semantic and functional equivalence of $x$, while reducing token length.

We will also present the technical details of the two prompt compression methods we employed.

**SELECTIVE-CONTEXT** Li et al. (2023) ranks and filters lexical units (e.g., tokens, phrases, or sentences) based on their informativeness. Informativeness is measured using *self-information*, defined for a token $x_t$ as:

$$I(x_t) = -\log_2 P(x_t | x_0, \dots, x_{t-1}). \tag{14}$$

In practice, self-information is calculated with smaller models for efficiency. Tokens with higher self-information are considered more informative, while redundant tokens have lower scores. To avoid disjoint filtering, tokens are grouped into larger lexical units (e.g., noun phrases or sentences). The self-information of each unit is computed by summing the scores of its tokens. Units are ranked by their scores, and a percentile-based threshold of $p$ is applied to retain the most informative content.

**LONGLLMLINGUA** Jiang et al. (2024) aligns closely with RAG use cases, decomposing prompt compression into modular steps:

- **Coarse-Grained Compression:** Documents are ranked by relevance using perplexity conditioned on the question: $r_k = -\frac{1}{N_c} \sum_{i=1}^{N_c} \log p(x_i^{\text{que, restrict}} | \mathbf{x}_k^{\text{doc}})$, where higher $r_k$ values prioritize relevant documents.

- **Fine-Grained Compression:** Token-level relevance is evaluated with *contrastive perplexity*: $s_i = \text{perplexity}(x_i | x_{<i}) - \text{perplexity}(x_i | x^{\text{que}}, x_{<i})$, highlighting critical tokens based on their importance to the query.

- **Adaptive Compression Ratio:** Compression budgets are dynamically allocated using: $\tau_k^{\text{doc}} = \max\left(\min\left(\left(1 - \frac{2I(r_k)}{K'}\right)\delta\tau + \tau^{\text{doc}}, 1\right), 0\right)$, where higher-ranked documents ($I(r_k)$) receive lower compression ratios.

- **Subsequence Recovery:** Ensures content integrity by 1) identifying the longest matching substring $\widetilde{\boldsymbol{y}}_{\text{key},l}$ in the LLM's response, 2) matching it with the maximum common subsequence $\boldsymbol{x}_{i,j}$ in the original prompt, and 3) replacing response tokens with the original prompt's subsequence.

- **Optimization Objective:** The overall objective balances output accuracy and compression: $\min_{\widetilde{\mathbf{x}}} D_\phi\left(\mathbf{y}, \widetilde{\mathbf{y}}\right) + \lambda\|\widetilde{\mathbf{x}}\|_0$, where $D_\phi$ measures the divergence between the original and compressed prompts' outputs, and $\lambda$ controls the compression tradeoff.

This approach ensures compressed prompts remain concise and informative, optimizing both efficiency and effectiveness for long-context scenarios.

## D.2 RERANKERS

To improve the precision and relevance of retrieved results, our pipeline employs rerankers to reorder coarse retrieval outputs. Below, we describe the underlying mechanisms of rerankers and their role in our system. To ensure clarity, we also briefly outline the retriever's principle and contrast it with the reranker.

**Retriever.** We only consider dense retrieval here. The retriever uses a **dual-tower architecture**, wherein:

- **Query Encoder**: Encodes the query into a dense embedding $\mathbf{q} \in \mathbb{R}^d$.

- **Document Encoder**: Encodes each document into a corresponding dense embedding $\mathbf{d} \in \mathbb{R}^d$.

The similarity between a query $\mathbf{q}$ and a document $\mathbf{d}_i$ is computed using a dot product:

$$\text{score}(\mathbf{q}, \mathbf{d}_i) = \mathbf{q}^\top \mathbf{d}_i, \quad i = 1, \ldots, N. \tag{15}$$

The retriever selects the top-$k$ documents with the highest scores as candidates. This coarse retrieval process is efficient and scalable because document embeddings can be pre-computed independently and stored, allowing for rapid approximate nearest neighbor (ANN) searches in vector space Douze et al. (2024); Johnson et al. (2019); Malkov & Yashunin (2018), which is ideal for large-scale retrieval. However, this independence of query and document encoding also makes the retriever less sensitive to context, as it cannot fully capture the nuanced interactions between queries and documents.

**Reranker.** Rerankers are employed to refine the results of coarse retrieval by reordering and filtering the candidate documents based on relevance. To overcome the drawbacks stated above, rerankers use a **cross-encoder architecture** to jointly encode the query and document, capturing their semantic interactions.

The reranker operates as follows:

- **Input Preparation**: Each query-document pair $(q, d_i)$ is concatenated into a single sequence, i.e.: $\{\texttt{[CLS]}, q, \texttt{[SEP]}, d_i, \texttt{[SEP]}\}$, where $\texttt{[CLS]}$ and $\texttt{[SEP]}$ are special tokens for encoding in Transformer-based models. This is a typical setup for cross-encoder architectures.

- **Contextual Encoding**: The concatenated sequence is input into a transformer (e.g., BERT), which computes a joint representation of the query and document. This step enables the model to capture rich contextual interactions, which are absent in retrievers due to their independent encoding process.

- **Relevance Scoring**: A relevance score is computed to quantify the alignment between the query and the document. In a standard cross-encoder setup, the output corresponding to the $\texttt{[CLS]}$ token is passed through a scoring head (e.g., a linear layer):

$$\text{score}(q, d_i) = f\left(\mathbf{h}_{\texttt{[CLS]}}\right), \tag{16}$$

where $\mathbf{h}_{\texttt{[CLS]}}$ represents the contextual representation of the $\texttt{[CLS]}$ token. Alternatively, some architectures may use pooling methods (e.g., mean or max pooling) over all token representations or token-level interactions to derive the relevance score.

- **Reordering and Selection**: Based on the computed relevance scores, the candidate documents are reordered, and the top-$k$ items are selected for downstream processing.

The key differences between retrievers and rerankers are summarized in Table 13. While retrievers are efficient and suitable for coarse retrieval over large document collections, rerankers excel in precision by capturing query-document interactions.

| Module | Retriever | Reranker |
|---|---|---|
| **Architecture** | Dual-tower (independent ) | Cross-encoder (joint ) |
| **Input** | Separate query and document inputs | Concatenated query-document pair |
| **Output** | Dot-product score for similarity | Relevance score for each pair |
| **Efficiency** | High efficiency, scalable to corpus | Costly, only for candidate sets |
| **Interaction** | No interaction between pairs | Captures rich semantic interactions |
| **Use Case** | Coarse-grained candidate selection | Fine-grained reordering and filtering |

Table 13: Comparison between retriever and reranker mechanisms.

### D.3 DATASET SELECTION

We select commonly used Single QA and Multi-hop QA datasets for inference to evaluate the performance of *SePer* in different scenarios. The dataset selection is guided by the need to cover a variety of QA tasks, ensuring a more comprehensive evaluation. Wherever possible, we perform inference on the $\texttt{test}$ set; if the $\texttt{test}$ set is unavailable, we use the $\texttt{dev}$ set instead. We re-sample the datasets, and for datasets with more than 1000 instances, we randomly select 1000 examples for inference. Figure 3 in the appendix presents the basic information of our utilized datasets.

### D.4 HYPERPARAMETER SETTING

We conduct experiments using the Llama 2 model series Touvron et al. (2023) from the Meta Llama family, specifically $\texttt{Llama-2-7b-chat-hf}$[1], $\texttt{Llama-2-13b-chat-hf}$[2], and $\texttt{Llama-2-70b-chat-hf}$[3]. Considering that the task involves instruction-following generation, we choose the chat versions of these models. To generate various and complete answers of various kinds for *SePer* computation, we set the temperature parameter of each model to 1.0, enabled $\texttt{do\_sample}$, and set the maximum tokens for generation to 512. For the retrieval corpus, we use the DPR version of the Wikipedia December 2018 dataset[5] as our retrieval corpus, following the configuration we utilize in the RAG framework FlashRAG Jin et al. (2024). We experiment with the set of top-$k$ values for retrieval being $\{1, 5, 10\}$, and follow each method's official implementation for the hyper-parameters of different prompt compression methods. For reranker usage, we set the reranker model as $\texttt{BAAI/bge-reranker-large}$[4]. We set the initial top-$k$ value for retrieval to 20 and then apply the set as $\{1, 5, 10\}$ for the reranker to choose items, leveraging the reranker's ability to both rank and filter out irrelevant content. We enable half precision when calculating *SePer*.

---

[1] https://huggingface.co/meta-llama/Llama-2-7b-chat-hf
[2] https://huggingface.co/meta-llama/Llama-2-13b-chat-hf
[3] https://huggingface.co/meta-llama/Llama-2-70b-chat-hf
[4] https://huggingface.co/BAAI/bge-reranker-large
[5] https://archive.org/download/enwiki-20181220

### D.5 PROMPT DESIGN AND IMPLEMENTATION

The selection of prompts is crucial for enabling large language models (LLMs) to understand tasks and produce responses that align with the desired style and requirements. In this work, we present two types of prompts: those that generate responses directly without retrieval and those that include references for retrieval-augmented generation. Specifically, we leverage the prompts introduced in Jin et al. (2024), which are listed as follows:

---

**Prompt for naive generation**

Answer the question based on your own knowledge. Only give me the answer and do not output any other words.
Question: {*question*}

---

**Prompt for RAG**

Answer the question based on the given document. Only give me the answer and do not output any other words.
The following are given documents.{*reference*}
Question: {*question*}

---

### D.6 SYSTEM SPECIFICATIONS FOR REPRODUCTIVITY

Our experiments were conducted on high-performance servers, each equipped with either an Intel(R) Xeon(R) Platinum 8378A CPU @ 3.00GHz or an Intel(R) Xeon(R) Platinum 8358P CPU @ 2.60GHz, 1TB of RAM, and 4/6 NVIDIA A800 GPUs with 80GB memory. Machines with 4 GPUs are configured with the SXM4 version, while those with 6 GPUs use the PCIe version. The software environment included *Python* 3.11, *PyTorch* 2.4, and *NCCL* 2.21.5 for reproductivity.

