# OpenReview forum: "SePer: Measure Retrieval Utility Through The Lens Of Semantic Perplexity Reduction"
_ICLR.cc/2025/Conference — ICLR 2025 Spotlight_

### Official Review · Reviewer_FzvM · 2024-11-01

**Soundness:** 3
**Presentation:** 3
**Contribution:** 3
**Rating:** 5
**Confidence:** 3

**Summary:**

This paper proposes SePer, an automatic evaluation metric to measure the benefit of retrieval. SePer first samples multiple responses of the input query, clusters them based on the semantic meanings, and estimates the belief distribution conditioned on the current input. It computes the CE between this belief distribution and ground truth as the final information gain. Experimental results demonstrate that SePer can evaluate retrieval utility and align with human preference.

**Strengths:**

- The method is well-formulated and written.

- SePer has a high correlation with human annotations, which is valuable for content evaluation.

**Weaknesses:**

- The assumption that retrieval utility is uniformly distributed across different reasoning steps may not hold true universally, potentially oversimplifying complex retrieval interactions.

- The need for extensive Monte Carlo sampling and clustering in semantic space may not be feasible for all applications, particularly those with limited resources.

- My major concern is about the experiment section. The selected metrics are evaluating semantic similarity instead of information gain or retrieval utility. Comparing SePer with the baselines does not necessarily mean that it is more competitive in measuring knowledge gain. Answers with high certainty but poor information (e.g., "I don't know") can still achieve high metric scores. The experimental results demonstrate that SePer is more about answer quality than information gain.

**Questions:**

- How much does it cost to run SePer during evaluation? The introduction claims that SePer is efficient but there isn't any cost analysis.

---

> ### Author Response · Authors · 2024-11-22
>
> Thanks for appreciating the clear formulation and good performance of the proposed *SePer* metric for evaluating retrieval utility.  We believe your concerns can be addressed through clarification and added experiments as follows:
>
>
>
> > 1.  "The assumption that retrieval utility is uniformly distributed across different reasoning steps may not hold true universally"
>
> We totally understand your concern, which is reasonable in a general view. However,  actually this assumption can hold for current multi-step reasoning datasets, including those in our papers. The reason lies in the construction process of these synthetic datasets. For example, the 2wikimultihopQA dataset constructs a multi-hop reasoning question by chaining two single-hop questions together. In this construction setting, it is reasonable to assume that each hop's retrieval contributes evenly to final success.  Besides, to the best of our knowledge, this is the only feasible way to obtain ground-truth middle-step retrieval utility annotations currently. We'd be happy to test for future datasets with more fine-grained annotations.
>
> We believe that the proposed *SePer* metric works in general multi-hop reasoning tasks by manually examining real cases. We **added several qualitative cases in the Appendix A.2.1** to prove that qualitatively for your reference.
>
>
>
> > 2. "Extensive Monte Carlo sampling and clustering in SePer calculation may be costly"
>
> The main cost of calculating SePer did come from sampling the generation 10 times. The clustering process (distance calculation) is actually ignorable compared to the sampling time since it is based on deBERTa. However, since the sampling choice is made following previous works like [1] in the uncertainty estimation area, it is inevitable if we want to achieve human-level and fine-grained evaluation. If more efficient and accurate uncertainty estimation algorithm emerges, they can be plugged in seamlessly as well. We also **added a Table in the appendix to show the time cost to compute SePer** compared to other LLM-as-a-judge methods marked in purple color. Since SePer is accurate, free from API, and explainable in computation, we believe there still exists many usage scenarios for SePer in spite of the extra cost brought by sampling 10 times.
>
>
>
> > 3. [Major concern] "whether SePer is more about answer quality than information gain"
>
> Thanks for bringing up this issue. We acknowledge that our writing has some ambiguity, especially between SePer and the reduction of SePer, which leads to some misunderstandings and confusion. We **distinguished the usage of these two terms by marking SePer reduction as Δ SePer to ensure consistency in the paper and add illustrative content**.  Here is the explanation FYI.
>
> - SePer (semantic perplexity) is a measure of the generator's confidence in the ground-truth answer, thus indeed more of a measure of answer quality as mentioned by the reviewer.
> - SePer reduction, computed from the difference of SePer w/ and w/o retrieval, measures the information gain acquired from retrieval, thus measuring "retrieval utility", as illustrated in the title and Figure 2.
>
> Thus, for the first concern, "Comparing SePer with the baselines" in Table 1 actually aims at proving *SePer* is a more fine-grained and accurate measure of LLM's knowledge on the correct answer, thus being a solid stone for calculating knowledge gain. Table 2 is the one that proves SePer reduction is a good measure of knowledge gain. For the second concern, "the Answers with high certainty but poor information (e.g., "I don't know")" won't have a high *SePer* score since we use the confidence assigned to the ground-truth answer (but not the confidence in the generated answer) for calculating SePer, as illustrated in Figure 2.
>
>
>
> > 4. How much does it cost to run SePer during evaluation？
>
> We **added the cost in time and money in Appendix A.2.2**, marked in purple color.
>
>
>
> Finally, thank you for your review and valuable comments. We appreciate your feedback and welcome any additional questions or suggestions.
>
>
>
> [1] Kuhn, Lorenz, Yarin Gal, and Sebastian Farquhar. "Semantic Uncertainty: Linguistic Invariances for Uncertainty Estimation in Natural Language Generation." *The Eleventh International Conference on Learning Representations*.

---

### Official Review · Reviewer_Ls69 · 2024-11-04

**Soundness:** 3
**Presentation:** 3
**Contribution:** 4
**Rating:** 8
**Confidence:** 3

**Summary:**

This paper proposes a new measure, SePer, to measure the retrieval utility through the lens of semantic perplexity reduction.  The authors point out the current problems at evaluating RAG effectiveness: assessing retrieval and generation components jointly, which obscures retrievals distinct contribution, or examining retrievers using traditional metrics such as NDCG. To address these limitations, the authors introduce an automatic evaluation method that measures retrieval quality through the lens of information gain within the RAG framework, by proposing Semantic Perplexity (SePer) that captures the LLM’s internal belief about the correctness of the retrieved information. The measure quantifies the utility of retrieval by the extent to which it reduces semantic perplexity post-retrieval. Extensive experiments demonstrate that SePer not only aligns closely with human preferences but also offers a more precise and efficient evaluation of retrieval utility across diverse RAG scenarios.

**Strengths:**

1.	Overall, the motivation of the paper is justified, and the paper is clearly written.

2.	The idea of introducing a new measure, SePer, to evaluate the effectiveness of retrieval in RAG is justified and it seems to be necessary and innovative.

3.	Evaluation of retrieval utility based on shifts in LLMs’ knowledge distributions sounds reasonable.

4.	The experiments show that the new measure not only aligns closer with human annotations but also is more consistent with inference-time situations.

5.	Both theoretical analysis and extensive experiments demonstrate that SePer provides a more accurate, fine-grained, and efficient evaluation of retrieval utility. It is also generalizable across a broad range of RAG scenarios.

**Weaknesses:**

1.	The design is mainly based on the entailment model and semantic clustering.  Moreover, the semantic clustering is also based on the entailment threshold, as show in its algorithm: Grouping responses rj into clusters C = {Ck} such that E(ri, rj) ≥ τ within each cluster.  Thus, the entailment model becomes the most critical factor in this measurement.  I am not sure if further discussion is needed with different entailment inference methods and different thresholds and see how they may impact the effectiveness of the SePer measure.  I feel some discussion is needed.

2.	The design does not conduct any concrete semantic analysis on different kinds of queries and texts to be retrieved.   It seems such study and discussion could be useful to deepen the understanding although this study (simply) tested on multiple datasets as shown in Table 2.

3.	There are quite a few typos in the paper.  For example, “a efficient” should be “an efficient” on lines 349 and 404;line 351: “It’s high correlation score in matching responses and answers also set a solid step …” should be  “Its high correlation scores in matching responses and answers also set a solid step …”; line 399 “results are shown in 3” should be “results are shown in Figure 3”, and others.

**Questions:**

The points outlined in the “weakness” should be clarified in the rebuttal.

---

> ### Author Response · Authors · 2024-11-22
>
> We appreciate the reviewer's valuable suggestions. We thank the reviewer for recognizing our work's clear motivation and presentation, as well as the necessity and innovation of SePer for evaluating retrieval in RAG. We are glad to find that the effectiveness, efficiency, and generalizability across RAG scenarios were highlighted. We believe your concerns can be addressed through clarification and additional details in our responses below.
>
>
>
> > W1: " How different entailment inference methods and different thresholds impact SePer measure "
>
> Thanks for this valuable suggestion. We are adding more ablation experiments in the appendix about SePer's robustness to hyperparameters, and this part will be marked in yellow as we push the revision in 1-2 days.  For now, we can say that $SePer_{soft}$ does not bear the problem of choosing a threshold since it directly incorporates the entailment score in weighted sum computation.
>
>
>
> > W2: "...conduct more concrete semantic analysis of cases to deepen the understanding of SePer".
>
> Thanks for this great suggestion. We do agree that concrete and qualitative case studies can facilitate the understanding of the characteristics of SePer such as its fine-grained evaluation ability. We **add several case studies in appendix A.2.1**, covering both simple/reasoning-type of questions and known/unknown questions to LLM.
>
>
>
> > W3:  "quite a few typos in the paper"
>
> Thanks a lot for your careful review. We have modified these typos and proofread our paper again to avoid typos.
>
>
>
> Finally, thank you again for your review and valuable comments. We appreciate your feedback and welcome any further questions or suggestions.

---

> > ### Author Response · Authors · 2024-11-25
> > **[Update] Experiments regarding W1 added**
> >
> > To address W1, we added extended experiment results in Appendix A.2.4 (marked in yellow) and Figure 7, please check out.

---

> > > ### Comment · Reviewer_Ls69 · 2024-11-26
> > >
> > > The authors have answered my questions convincingly.   I have no more concern on it.  I will maintain my positive evaluation score on this paper.

---

### Official Review · Reviewer_E3Wb · 2024-11-04

**Soundness:** 1
**Presentation:** 1
**Contribution:** 1
**Rating:** 8
**Confidence:** 2

**Summary:**

The authors propose a method for estimating the value of a document in a RAG setting according to its ability to reduce semantic perplexity in the downstream model. The method is compared to ground truth utility and human preferences using measured correlation.  The method is further studied in the context as an evaluation metric.

**Strengths:**

* **Interesting new work in relevance estimation.** Determining the relevance of an item for RAG is an exciting new area of research that moves beyond classical notions of relevance from information retrieval.

**Weaknesses:**

* **Metric validation.** When proposing a new metric, various criteria must be met. These include validity (e.g., construct, content, convergent, discriminate), reliability, and sensitivity.  Although Section 4.1 provides some evidence of validity, we need more analysis to understand if this would be a strong metric (e.g., reliability, sensitivity).  The remaining experiments in Section 5 are difficult to connect and are relatively informal.  The authors are strongly encouraged to review the literature in metric design, some of which are listed below,
   * Abigail Z. Jacobs and Hanna Wallach. Measurement and fairness. In Proceedings of the 2021 acm conference on fairness, accountability, and transparency, FAccT '21, 375--385, New York, NY, USA, 2021. , Association for Computing Machinery.
   * Claudia Wagner, Markus Strohmaier, Alexandra Olteanu, Emre Kıcıman, Noshir Contractor, and Tina Eliassi-Rad. Measuring algorithmically infused societies. Nature, 595(7866):197--204, 2021.
   * Ziang Xiao, Susu Zhang, Vivian Lai, and Q. Vera Liao. Evaluating evaluation metrics: a framework for analyzing NLG evaluation metrics using measurement theory. In Houda Bouamor, Juan Pino, and Kalika Bali, editors, Proceedings of the 2023 conference on empirical methods in natural language processing, 10967--10982, Singapore, December 2023. , Association for Computational Linguistics.
* **Related work in information retrieval.** The authors seem to dismiss much of the information retrieval work that looks at the cognitive process of searching and its connection to relevance.  Since they do cite earlier work (e..g, [Cooper 1973]), I believe this should be considered.  Please look into work from Belkin and related,
   * Nicholas Belkin. Anomalous states of knowledge as a basis for information retrieval. Canadian Journal of Information Science, page 133-143, 11 1980.
   * N. J. Belkin, R. N. Oddy, and H. M. Brooks. Ask for information retrieval: part i. background and theory. Journal of Documentation, 38(2):61-71, June 1982.
   * N. J. Belkin, R. N. Oddy, and H. M. Brooks. Ask for information retrieval: part ii. results of a design study.. Journal of Documentation, 38(3):145-64, September 1982.
* **Writing.** This is minor but the authors should consider improving the flow and exposition in the writing.  There seems to be little justification for the Properties in Section 3.2, which could benefit from discussion or citation.  The RQs in Section 5 are introduced also without much discussion of why they are important.

**Questions:**

The authors have addressed the above weaknesses.

---

> ### Author Response · Authors · 2024-11-22
>
> It is interesting to link our work to redefining relevance in IR. We believe that this connection may occur due to the reviewer's familiarity with the anomalous state of knowledge (ASK) theory, which proposed to improve IR by mining users' knowledge gap. However, we must clarify here that despite sharing some cognitive similarities, the goals, motivations, and scenarios are all shifted from this line of work. In short, although the reviewer put it in the Strengths part that "Determining the relevance of an item for RAG is exciting", we state that our work does not focus on redefining "relevance" to improve IR, but to propose a computable metric to evaluate the "retrieval utility" in RAG. The in-depth discussion of nuances between them is put at the end of a rebuttal for anyone interested in this topic.
>
> For other concrete questions, we address them as follows:

---

> ### Author Response · Authors · 2024-11-22
>
> 1. **Metric validation**.
>
>     We believe there is some misunderstanding regarding this point. We totally agree that a new proposed metric should be tested upon its reliability and validity as mentioned by reference [1] from the reviewer and we believe we had implicitly followed the scientific convention of proposing a new metric in the following part of this paper:
>       1) **Reliability**: The reliability of a metric measures its sensitivity against random measurement errors, such as consistency across repeated measures, and choice of datasets in a benchmark [1].
>           1. For the randomness due to repeated sampling, we conducted experiments in Figure 3 in the original paper. In Figure 3, we repeatedly sampled the n generations 10 times at each fixed n and calculated the variance of correlation for repeated measurements. The shadow area in Figure 3 quantifies the error range of our SePer metric. Shortly speaking, there are two conclusions to take away from the experiments in Figure 3:
>               1. The **Coefficient of Variation of correlation is within 1%** using our default hyperparameters as proven by experiments from four major datasets, indicating high reliability and stability of our SePer metric in aligning with human preferences. (1% generally means high precision measurements in common convention )
>               2.  **As n (the number of generations) increases, the variance generally decreases even smaller** as shown by the length of the error bar in Figure 3. Until our default setting of n=10, the error range is small enough to qualify *SePer* as a reliable metric, striking a good balance between accuracy and efficiency.
>
>               To highlight this point as suggested by the reviewer, **we further add a Table in Appendix A1.1.2 to illustrate the implementation details of the test of robustness.** We also add statistics of standard deviation and coefficient of computing SePer reduction to prove its stability in computation in A1.1.2.
>           2. For randomness of datasets, we conducted experiments on major mainstream RAG datasets, covering aspects of simple tasks (NQ, TriviaQA, etc.) and reasoning-based tasks (Hotpot QA, etc.). The results of the proposed SePer consistently align well with human preferences. To add further evidence, **we add benchmarks of applying SePer to retriever evaluation on different datasets in Appendix,** where we can see from the produced ranks that the proposed SePer is robust and consistent across different datasets.
>
>
>       2) **Validity**: the validity of a metric measures whether it reasonably reflects the intended construction purpose and aligns well with human preferences[1]. To prove *SePer*'s validity, we combine theoretical along with experimental analysis. Originated from cognitive perspectives [4][5], the intuition is widely-accepted.  Grounded in the Bayesian inference frameworks, we construct the metric with stepwise derivation as proved in section 3. As for experiments, we proved in Table 1 and Table 2 that the proposed metric aligns better with human preferences across different datasets. **The P-values in hypothesis tests are all less than 0.01, indicating a significant correlation relationship with human judgments.** We added more detailed captions to explain these experiments.
>
>       In all, we believe that our experiments satisfy the required justification of metric design. We follow the writing style of related works such as bertscore [2] and factscore [3] in this field and have at least the amount of workloads on par with these works.
>
>       However, we do agree with the reviewer that the justification of metric validation is critical in a metric design paper and should be highlighted more clearly. To this end, we reorganized the writing of the corresponding part (section 4) based on the taxonomy of [1] mentioned by the reviewer and added more details in the appendix.
>
>
> 2. **Related works**.
>
>     Thanks. we agree that the premise in the anomalous state of knowledge(ASK) theory shares some similar ideas with our intuition of measuring the success of retrieval by knowledge gain (the "anomaly" in ASK). We cited these works in the section 1 and 3 of our paper.
>
>
> 3. **Writing**.
>
>     Thanks for this suggestion. We acknowledge that some parts of writing in the original paper are too brief and make it prone to misunderstanding. We added a leading paragraph in section 5 for the background introduction and added citations in section 3 as suggested. We also re-organize section 4 to emphasize the justification of metric design.
>
>
>
> Finally, thank you for your review and valuable comments. We appreciate your feedback and welcome any additional questions or suggestions.

---

> > ### Comment · Reviewer_E3Wb · 2024-11-24
> >
> > 1. ***Metric validation***
> >
> > 1. ***Reliability:*** Thank you very much for addressing concerns of metric reliability.  This is good.
> > 2. ***Validity:***. Unfortunately, the connection between the older work from Cooper/Belkin, while welcome, is not thoroughly discussed.  Questions remaining to be answered are (a) what aspects of these theories support the properties, (b) what are alternative theories that were not considered. The alignment experiments are good but are missing any indication of what significance test was conducted or if there was correction for multiple comparisons.
> >
> > 2. ***Related works.*** I implore the authors to engage with the literature more than a citation.  There are clear connections (discussed below) that are important to place the work in scholarly context.
> >
> > 3. ***Writing.*** Thank you for improving this writing.

---

> > > ### Author Response · Authors · 2024-11-25
> > >
> > > Glad to hear that 2/4 of your concerns regarding the reliability of methods and writing are addressed after the previous discussion. We believe that your remaining questions can be addressed as follows:
> > > 1. **Validity**:
> > >
> > > > "...the connection between the older work from Cooper/Belkin, while welcome, is not thoroughly discussed"
> > >
> > > We **revised Section 3.2 and attached an explanation paragraph after each property** about how it is connected with / inspired by those cognitive IR works, including [1][2][3][4], highlighted in the color of maroon.
> > >
> > > > "The alignment experiments are good but are missing any indication of what significance test was conducted"
> > >
> > > We used $t$-test to assess the significance of the Pearson correlation, with $p$-value < 0.01 for all reported figures about SePer. For brevity, we **added details of the t-test in the Metrics part on Page 7, Section 4.1.2, highlighted in the color of maroon**. This should make the alignment experiments more complete and clear thanks to your suggestion.
> > >
> > > 2. **Related works**
> > >
> > > > "...engage with the literature more than a citation."
> > >
> > > Same as 1. we **added deeper and more concrete discussion about more related works along the lines of cognitive IR in Section 3.2 highlighted in color maroon**, discussing how the utility definition of our proposed metric can connect to previous cognitive science works. Please check it out.
> > >
> > >
> > > **References**:
> > >
> > > [1] William S Cooper. A definition of relevance for information retrieval. Information storage and
> > > retrieval, 7(1):19–37, 1971.
> > >
> > > [2] N. J. Belkin, R. N. Oddy, and H. M. Brooks. Ask for information retrieval: Part i. background and
> > > theory. Journal of documentation, 38(2):61–71, 1982.
> > >
> > > [3] Brenda Dervin. On studying information seeking methodologically: the implications of connecting
> > > metatheory to method. Information Processing & Management, 35(6):727–750, 1999.
> > >
> > > [4] Peter Ingwersen. Cognitive perspectives of information retrieval interaction: elements of a cognitive
> > > ir theory. Journal of documentation, 52(1):3–50, 1996.

---

> > > > ### Comment · Reviewer_E3Wb · 2024-11-25
> > > >
> > > > Thank you very much!
> > > >
> > > > Did you correct for multiple comparisons in your $t$-test?

---

> > > > > ### Author Response · Authors · 2024-11-26
> > > > >
> > > > > Our study belongs to the scope of **single hypothesis testing** and does not fall under the category of multiple comparisons. Here is a more detailed explanation:
> > > > >
> > > > > In our case, variable 1 is *SePer*, and variable 2 is human annotation. In contrast, multiple testing typically involves evaluating variable 1 against a *group* of variables, i.e., different dimensions on the same set of observed data samples. To further clarify this point, here are several examples from statistics textbooks where calibration for multiple comparisons is necessary:
> > > > >   1. To test the hypothesis "whether a treatment is effective on patients", the researchers do multiple hypothesis testing on multiple indicators of these patients, such as blood pressure and heart rate.
> > > > >   2. To test the hypothesis "whether high-voltage power lines impact residents' health", the researchers did multiple hypothesis testing on the occurrence of over 800 types of diseases on these residents.
> > > > >
> > > > > These scenarios involve hypotheses constructed based on a group-level granularity, necessitating extra calibration to avoid overestimating correlation.
> > > > >
> > > > > We understand the reviewer's concern about multiple comparisons lies in that we tested on 7 different datasets. However, this "multiple" division is along the data dimension rather than across parameters, factors, or attributes. Thus, the errors do not accumulate. This reasoning is consistent with why other works of a similar nature, such as BERTScore, do not perform multiple comparison calibration.

---

> > > > > > ### Comment · Reviewer_E3Wb · 2024-11-26
> > > > > >
> > > > > > There may be some confusion. It sounds likes you tested the significance of the correlation, which is good. However, you are also making a claims comparing metrics (eg for a fixed column across rows in table 1). For possible corrections see,
> > > > > >
> > > > > > Rotem Dror, Gili Baumer, Marina Bogomolov, and Roi Reichart. 2017. Replicability Analysis for Natural Language Processing: Testing Significance with Multiple Datasets. Transactions of the Association for Computational Linguistics, 5:471–486.

---

> > > > > > > ### Author Response · Authors · 2024-11-26
> > > > > > >
> > > > > > > We will check the details of your suggested paper during the extended discussion period within the next 1-2 days. However, given the pressing deadline for paper revisions, we need to first confirm that all previously discussed issues (except for this one about the multiple comparison correction of $t$-test) have been addressed, and would like to kindly inquire whether there are any other major concerns about the overall quality of the paper that remain unresolved, allowing us to make the necessary revisions before the deadline. Thanks!

---

> > > > > > > > ### Comment · Reviewer_E3Wb · 2024-11-26
> > > > > > > >
> > > > > > > > I appreciate all of your hard work in responding to the feedback. Yes, this is the one remaining issue.

---

> ### Author Response · Authors · 2024-11-22
>
> [1] Ziang Xiao, Susu Zhang, Vivian Lai, and Q. Vera Liao. Evaluating evaluation metrics: a framework for analyzing NLG evaluation metrics using measurement theory. In Houda Bouamor, Juan Pino, and Kalika Bali, editors, Proceedings of the 2023 conference on empirical methods in natural language processing, 10967--10982, Singapore, December 2023. , Association for Computational Linguistics.
>
> [2] Zhang T, Kishore V, Wu F, et al. BERTScore: Evaluating Text Generation with BERT[C]//International Conference on Learning Representations.
>
> [3] Sewon Min, Kalpesh Krishna, Xinxi Lyu, Mike Lewis, Wen-tau Yih, Pang Koh, Mohit Iyyer, Luke Zettlemoyer, and Hannaneh Hajishirzi. 2023. FActScore: Fine-grained Atomic Evaluation of Factual Precision in Long Form Text Generation. In *Proceedings of the 2023 Conference on Empirical Methods in Natural Language Processing*, pages 12076–12100, Singapore. Association for Computational Linguistics.
>
> [4] Cooper, William S. "On selecting a measure of retrieval effectiveness." *Journal of the American Society for Information Science* 24.2 (1973): 87-100.
>
> [5] Nicholas Belkin. Anomalous states of knowledge as a basis for information retrieval. Canadian Journal of Information Science, page 133-143, 11 1980.

---

> ### Author Response · Authors · 2024-11-22
> **Some extended discussion on the topic of ASK (Anomalous State of Knowledge) theory**
>
> We would like to thank the reviewer for bringing up the ASK theory. We added some thoughts and discussions about the difference between our work and ASK theory after we went through related works suggested by the reviewer. Through this public discussion, we hope to facilitate the understanding of our paper, as well as encourage new insights along the lines of the interesting work in the 1980's.
>
> We find resonance with ASK theory in the assumption that "knowledge gain (bridging knowledge gap) is the key to the success of information retrieval". However, we actually work in different directions, as illustrated in the following aspects:
>
> - **Goals**: ASK aims at improving the design of IR **methods**, while our proposed SePer aims at improving the **evaluation** of retrieval utility in recent RAG pipelines.
> - **Subjects**: In ASK, a real human is the information recipient, while in our evaluation framework, LLM is the information recipient. This results in different methods we can leverage for computation. For example, it's still infeasible to quantitatively estimate belief shifts in human brain, but it's possible to estimate LLM's internal belief in a computable way as described in this paper, thanks to recent research about LLMs and uncertainty estimation.
> - **Motivations**: ASK is motivated by the fact that users often cannot express their real information needs during searching. Thus,  it's important for the search engine to identify the lack of knowledge (anomalous state of knowledge) in a specific user, which is underrepresented but critical to satisfying the real information needs of the user. To this end, ASK proposed a new perspective of retrieving documents that can bridge the knowledge gap rather than that can match the expressed knowledge as done by traditional methods such as BM25. While we appreciate the importance and keenness of this perspective, we'd like to kindly state that our motivation stems primarily from a pragmatic need to address the limitation of evaluation in recent RAG pipelines.
> - **Scenarios**: ASK is especially useful in cases when the user cannot fully express their needs. In our setting, our methods do not focus on the gap between "know" and "tell".
> - **Methods**: ASK focuses on picturing users' state of knowledge and detecting the anomalous state using methods like interactive querying, and then improving retrieval accuracy accordingly. We focus on estimating LLM's belief through a Bayesian inference framework via sampling and clustering, and then use it to evaluate retrieval utility.
>
> We welcome and appreciate further discussions on related topics.

---

> > ### Comment · Reviewer_E3Wb · 2024-11-24
> >
> > Thank you for your thorough review of ASK as it pertains to RAG.
> >
> > I would encourage the authors not only engage with the original ASK paper, but the larger body of work on cognition and search, especially as the motivations and methods are very close (see, e.g., "search as learning").  This is especially important as you now include Belkin and Cooper in your motivation for the properties in Section 3.2 _without any discussion as to the connection_.  I strongly encourage the authors to more explicitly draw out how these references support the properties since, otherwise, it's unclear, especially for readers unfamiliar with the methods.
> >
> > With respect to the differences between the submissions and ASK, I first encourage the authors not to fixate too much on ASK but to look at the broader literature on search and cognition.  To the specific points,
> >
> > > ASK aims at improving the design of IR methods, while our proposed SePer aims at improving the evaluation of retrieval utility in recent RAG pipelines.
> >
> > This distinction is a little artificial since the objective of evaluation is improving methods (Section 5).
> >
> >  > In ASK, a real human is the information recipient, while in our evaluation framework, LLM is the information recipient. This results in different methods we can leverage for computation. For example, it's still infeasible to quantitatively estimate belief shifts in human brain, but it's possible to estimate LLM's internal belief in a computable way as described in this paper, thanks to recent research about LLMs and uncertainty estimation.
> >
> > There are certainly real differences between a person and a machine engaging with a search system, but quantification may not be the largest.  Work in education and "search as learning" captures this.
> >
> > > ASK is especially useful in cases when the user cannot fully express their needs. In our setting, our methods do not focus on the gap between "know" and "tell".
> >
> > This is an interesting distinction and one that the authors should explore in comparing with the broader literature on search and cognition.
> >
> > > ASK focuses on picturing users' state of knowledge and detecting the anomalous state using methods like interactive querying, and then improving retrieval accuracy accordingly. We focus on estimating LLM's belief through a Bayesian inference framework via sampling and clustering, and then use it to evaluate retrieval utility.
> >
> > These are indeed important methodological differences that should be highlighted.
> >
> > This whole line of feedback is more about (a) providing scholarly context for the new method (which can still be improved) and (b) providing support for the properties in Section 3.2 (which can be substantially improved).

---

> ### Author Response · Authors · 2024-11-27
>
> There is indeed some misunderstanding due to the gap between different subareas, mainly regarding the datasets.
>
> Here are the detailed explanations:
>
> Reference [1], while named as "multiple datasets", is implicitly based on the dimension of "attributes" or "aspects", each "aspect" is represented by a dataset, such as a language (multilingual setting) or a domain (multi-domain setting). In these cases, correction by the number of datasets indeed improves the soundness of replicative analysis, since the hypothesis is based on a composition of all aspects. For example, in the task of multilingual POS tagging, the p-value should be calibrated if one wants to solidly claim that "method A is better than method B in pos tagging of all languages", since the erroneous discovery on any dataset(language) would hurt the integrity of the global hypothesis.
>
> As shown by the experiments in [1], many traditional NLP tasks fall under the scope of "correction of multiple datasets because:
>
> 1. the hypothesis is claimed in a multi-aspect way such as multilingual and multi-domain, as illustrated above.
> 2. each test dataset is relatively small(from 280 to 2327 in [1], for example), thus p-values suffer from large variations.
>
> On the contrary, our "multiple datasets" do not represent different aspects, but rather a cumulation of data samples. This is usually the case in IR and QA area, since it's hard to divide these samples by aspects.
>
> To wrap up the difference between the two mentions of "multiple datasets",
>
> - The more hypotheses, the more prone to type-1 error, thus need calibration by methods mentioned in [1]
> - The more data samples, the less prone to type-1 error, reflected in the computation of $t$-value by normalizing by $n$.
>
>
> To make things more clear, we add a table here of the exact $p$-value of each dataset. The significance claim of $p$<0.01 still holds even if we do multiple corrections since the p-value is too small due to the large number of samples.
>
> ---
> |          | NQ | MSMARCO | SQuAD              | HotPotQA | 2WikimultihopQA | Musique | IIRC                |
> |----------|----|---------|--------------------|----------|-----------------|---------|--------------------|
> | $p$-value  | 0  | 0       | 2.86$\times e^{-207}$         | 0        | 0               | 0       | 2.88$\times e^{-218}$         |
> ---
> Note: the 0 $p$-value does not mean exactly equal to 0, but below the numerical precision between the difference of computed $p$ and 0.
>
> Nevertheless, thanks for bringing up the paper [1], which facilitates our thinking regarding the evaluation of NLP tasks.
>
> References:
>
> [1] Rotem Dror, Gili Baumer, Marina Bogomolov, and Roi Reichart. 2017. Replicability Analysis for Natural Language Processing: Testing Significance with Multiple Datasets. Transactions of the Association for Computational Linguistics, 5:471–486.

---

> > ### Comment · Reviewer_E3Wb · 2024-11-27
> >
> > I apologize but I am now more confused about the results.  So far in this discussion, the authors have described single $p$-value, and included a description in Section 4.1.2.  Let me go through my thinking for this part of my concern,
> >
> > ***Table 1***
> > * The hypothesis being tested is "SePer achieves the highest accuracy." I am interpreting that as "largest in the column."
> > * We need to understand if that is statistically significant.  Does that mean that, for each column, you are conducting six tests each between the row labeled _SePer_ and one of the six others (Lexical Match,..., Human)?  If not, can you please describe how you tested the hypothesis that "SePer achieves the highest accuracy"?
> > * Cells are described as correlations but then described as $F_1$ or accuracy values.  This is minor but you should just probably not use correlation in this situation.
> >
> > ***Table 2***
> > * The hypotheses tested are "∆SePerH and ∆SePerS significantly outperform other baselines in measuring retrieval utility."  I am interpreting this as two sets of nine comparisons for each column.
> > * It's not clear how a single $p$-value is computed for each column.
> > * I suspect—though I'm not sure—that the $p$-value reported above is for the correlation itself, not the _comparison of correlations_.

---

> > > ### Author Response · Authors · 2024-11-28
> > >
> > > Thanks for the stepwise clarification. We think that the current confusion lies in our divergent definitions of the hypothesis, thus the reported p-value is different from what you mean. Our hypothesis is that "SePer has a significant correlation with human judgments". We suspect that your assumed hypothesis is "SePer is better than baseline A (B,C,D...) in all datasets".
> > > Here we answer your concrete questions step by step:
> > >
> > > **Table 1**
> > > > The hypothesis being tested is "SePer achieves the highest accuracy." I am interpreting that as "largest in the column."
> > >
> > > Yes.
> > >
> > > > We need to understand if that is statistically significant. Does that mean that, for each column, you are conducting six tests each between the row labeled SePer and one of the six others (Lexical Match,..., Human)? If not, can you please describe how you tested the hypothesis that "SePer achieves the highest accuracy"?
> > >
> > > $t$-test is not conducted in Table 1, the evaluation process is following the original benchmark paper [1]. Sorry for causing this misunderstanding.
> > >
> > > > Cells are described as correlations but then described as or accuracy values. This is minor but you should just probably not use correlation in this situation.
> > >
> > > We use the word "correlation" and the metric "F1/ACC" to be consistent with the original benchmark paper [1].
> > >
> > > **Table 2**
> > >
> > > > The hypotheses tested are "∆SePerH and ∆SePerS significantly outperform other baselines in measuring retrieval utility." I am interpreting this as two sets of nine comparisons for each column.
> > >
> > > We did not conduct nine tests between each pair of methods. The correlation score is between ∆Seper(S,H) and human judgments. The hypothesis is "∆SePer has a significant correlation with human judgments.", and this is where the p-value is computed. "∆SePer outperforms other baselines" is not the hypothesis but the conclusion we draw from a higher correlation score with human judgments.
> > >
> > > > It's not clear how a single value is computed for each column.
> > >
> > > The provided p-value is on testing whether there is a significant correlation between ∆SeperS with human judgments.
> > >
> > > > I suspect—though I'm not sure—that the value reported above is for the correlation itself, not the comparison of correlations.
> > >
> > > Yes. You are correct.
> > >
> > > If the explanation provided aligns with your understanding, we suspect that we did not conduct the type of pairwise $t$-test that you were referring to.
> > >
> > > We agree that the way of pairwise $t$-test (as assumed by the reviewer) is the most rigorous and statistically sound way of proving "method A is better than B". However, we also hope to defend for ourselves in this review process that our approach of comparing methods by correlation score and $t$-test the significance of the correlation is consistent with the convention in recently published evaluation papers, such as BERTScore [1], ARES [2] and FIT [3].
> > >
> > > Nonetheless, we are willing to implement the kind of $t$-test suggested by the reviewer and [4], if the reviewer decided that this test would enhance the quality assessment of this paper. Thanks!
> > >
> > >
> > > **References**:
> > >
> > > [1] Zhang, Tianyi, et al. 2020. "BERTScore: Evaluating Text Generation with BERT." International Conference on Learning Representations.
> > >
> > > [2] Saad-Falcon, Jon, et al. "ARES: An Automated Evaluation Framework for Retrieval-Augmented Generation Systems." Proceedings of the 2024 Conference of the North American Chapter of the Association for Computational Linguistics: Human Language Technologies (Volume 1: Long Papers). 2024.
> > >
> > > [3] Zandonati, Ben, et al. 2023. "FIT: A Metric for Model Sensitivity." The Eleventh International Conference on Learning Representations.
> > >
> > > [4] Rotem Dror, Gili Baumer, Marina Bogomolov, and Roi Reichart. 2017. Replicability Analysis for Natural Language Processing: Testing Significance with Multiple Datasets. Transactions of the Association for Computational Linguistics, 5:471–486.

---

> > > > ### Author Response · Authors · 2024-11-28
> > > > **[Update] Pairwise $t$-test result**
> > > >
> > > > Update: We still implement the pairwise $t$-test because we think that it's the right thing to do. Despite its absence in recent literature, we agree with the reviewer that it is the most robust way to demonstrate that our method is better than the baselines. Here is the pairwise $t$-test result for your reference:
> > > >
> > > > |    | **NQ**      | **MSMARCO** | **SQuAD**   | **2WikiMHQA** | **HotpotQA** | **IIRC**    | **Musique** |
> > > > |---------------------|-------------|-------------|-------------|---------------|--------------|-------------|-------------|
> > > > | **Exact Match**     | 0           | 3.68E-260   | 1.51E-55    | 0             | 1.20E-265    | 1.51E-11    | 2.58E-115   |
> > > > | **ROUGE**           | 3.06E-161   | 3.03E-213   | 1.56E-23    | 7.28E-194     | 2.68E-53     | 3.70E-03    | 8.38E-32    |
> > > > | **BLEU**            | 0           | 2.53E-214   | 2.55E-57    | 0             | 0            | 6.86E-29    | 1.40E-162   |
> > > > | **BERTScore**       | 0           | 1.59E-292   | 2.64E-69    | 0             | 0            | 1.57E-40    | 2.94E-205   |
> > > > | **Perplexity**      | 2.80E-12    | 1.29E-13    | 3.80E-12    | 3.00E-11      | 7.13E-06     | 1.60E-11    | 1.10E-11    |
> > > > | **Entropy**         | 0           | 0           | 2.71E-109   | 0             | 0            | 0           | 0           |
> > > > | **Semantic Entropy**| 0           | 0           | 2.10E-124   | 0             | 0            | 0           | 0           |
> > > > | **Auto-J**          | 0           | 0           | 1.49E-116   | 0             | 0            | 0           | 0           |
> > > > | **Prometheus**      | 0           | 0           | 1.51E-111   | 0             | 0            | 0           | 0           |
> > > >
> > > > The above p-value is tested on the hypothesis that "$\Delta$ Seper_S is better than baseline X on dataset Y", tested pairwise. With the result above, we can make a more statistically sound claim that our proposed method is better than all other baselines, with 95% significance. The small p-values ensure that the conclusion still holds under correction.
> > > > Let us know if you have further suggestions.

---

### Official Review · Reviewer_tcfG · 2024-11-05

**Soundness:** 3
**Presentation:** 3
**Contribution:** 3
**Rating:** 8
**Confidence:** 2

**Summary:**

This paper proposes a measure of semantic perplexity reduction (SePer) to evaluate the utility of retrieved documents in RAG. The measure reflects the change in the probability of the ground truth answer when given the retrieved document. The authors evaluate the correlation of this measure with respect to machine and human judgments, showing higher correlation than other measures.SePer is then used to evaluate the effect of different factors: model size, number of retrieved document, prompt compression and document reranking.
The experiments suggest that SePer can be used to estimate whether a retrieved document is useful.

**Strengths:**

1. The paper uses a well motivated idea to measure the utility of a retrieved document. Similar idea has been used in IR literature quite a long time ago, but it is rarely used in recent studies. Reusing this idea with the current approaches to RAG is interesting.
2. The demonstration that SePer could be a reasonable measure for document utility is interesting (however, some unclarities - see weaknesses).

**Weaknesses:**

1. While the general idea is well motivated, its demonstration and tests have could be described more clearly. Some unclear presentation hampers the understanding. For example, the tables and figures are usually not well explained. E.g. In Table 1, it is unclear what the numbers correspond to. SePer reduction is a central concept of the paper. However, there is not a clear definition of it. One may guess that it corresponds to Equation 9 (?), but a clear explanation would help.
2. Some important details are not well described in the paper. For example, reranker is a means to improve RAG. However, it is unclear how reranking is done. The single sentence that explains it is unclear. The method used for prompt compression may also be more detailed.
3. The evaluation is limited to testing if SePer correlates to human judgments and how different factors influence SePer measure. It would be interesting to design a method to use SePer in RAG, e.g. to select retrieved documents according to SePer. This would be a more direct evaluation of the utility of SePer.
4. The paper bears some similarity to Farquhar et al. (2024). This has not been explicitly stated in the paper.

Note after rebuttal: The authors have addressed the above problems. More explanations are added, making the description clearer.

**Questions:**

1. How could SePer be used concretely in RAG?
2. SePer is compared to retrieval annotations to estimate its correlation to the latter. Would it be possible to test if SePer correlates well with the final answer generation, i.e. when a document with higher SePer is provided to the generation process, the generated answer tends to be better?

---

> ### Author Response · Authors · 2024-11-22
>
> We are grateful to the reviewer for recognizing the novelty and validity of the metrics we proposed. We appreciate your valuable suggestions and believe that your concerns can be addressed through clarification and additional details in our responses below.
>
>
>
> > W1&W2. "Some unclear presentation hampers the understanding"
>
> Thanks a lot for this suggestion. We do realize that there are some unclear writings that may confuse the readers. We **have added more detailed captions** to each figure and table. We also have **added an explanation about the concept of SePer reduction highlighted in red** and distinguish its use from SePer by marking it as Δ SePer.  As for explaining the role of reranking and prompt compression, we **added a leading paragraph in section 5 and a detailed tutorial about the current components in RAG in Appendix A.4**, which will help wider readers to understand section 5 about how to use SePer to measure the utility of different components in RAG.
>
>
>
> > W3&Q1. "How could SePer be used concretely in RAG?"
>
> Thanks a lot for this suggestion. We **ran experiments and added a table to benchmark current RAG workflows in Appendix A.3.2** and the procedure of applying SePer in RAG. We also agree that SePer can be used in tasks such as providing more fine-grained annotation in RAG datasets and will continue exploring more use cases in future research.
>
>
>
> >  W4. "Similarity to Farquhar et al. (2024) ... has not been explicitly stated"
>
> We actually cited the semantic entropy work from Farquhar et al. (2024) in section 1 and section 3.3, one for the ICLR version and one for the Nature version. We also **add some explanations of how our work builds upon and differs from this milestone work at the end of section 3.3**, marked in orange color.
>
>
>
> > Q2:  "if SePer correlates well with the the quality of final answer generation?"
>
> Yes, SePer correlates well with the answer generation quality. It measures the generator's belief on reference answer by design. Actually, Table 1 demonstrates whether the compute mechanism in SePer can better reflect answer quality, which is proven yes. Table 2 is about SePer reduction (not SePer), where we compute the correlation of SePer reduction to the retrieval utility and prove it is the best measure of retrieval utility. We apologize again for the confusing part in writing about SePer and SePer reduction. We examined thoroughly and **modified relevant parts to ensure the consistent usage of the two terms and mark SePer reduction as Δ SePer**. We also added an illustrative paragraph highlighted in red to help the reader clearly understand these two proposed concepts.
>
>
>
> Finally, thank you for your review and valuable comments. We appreciate your feedback and welcome any additional questions or suggestions.

---

### Meta-Review · Area_Chair_oiTn · 2024-12-24

**Metareview:**

This paper proposes a measure of semantic perplexity reduction (SePer) to evaluate the utility of retrieved documents in RAG. The measure reflects the change in the probability of the ground truth answer when given the retrieved document. SePer is used to evaluate the effect of different factors, e.g., model size, number of retrieved document. The authors have performed extensive experiments to demonstrate the effectiveness of SePer.

Overall, this paper is well motivated, and well written. The idea of the proposed method is novel. The experimental analysis sounds reasonable and convincing. Both theoretical analysis and extensive experiments demonstrate that SePer provides a more accurate, fine-grained, and efficient evaluation of retrieval utility.

**Additional Comments On Reviewer Discussion:**

In the rebuttal, the authors have included more descriptions about the proposed method. Some unclear descriptions have been well addressed in the revised version. Moreover, the authors have well addressed the reviewers' concerns regarding with how to use SePer for RAG, difference between this work and existing study Farquhar et al. (2024) , the correlation between SePer and the quality of the final answer, and the metric validation.

---

### Decision · Program_Chairs · 2025-01-22

Accept (Spotlight)